# Muscleblind-1 interacts with tubulin mRNAs to regulate the microtubule cytoskeleton in *C. elegans* mechanosensory neurons

Dharmendra Puri¤◉, Sunanda Sharma◉, Sarbani Samaddar, Sruthy Ravivarma◉, Sourav Banerjee, Anindya Ghosh-Roy◉*

National Brain Research Centre, Manesar, Gurgaon, Haryana, India

◉ These authors contributed equally to this work.
¤ Current address: Boston Children's Hospital, Harvard Medical School, HHMI, Boston, Massachusetts, United States of America
* anindya@nbrc.ac.in

**Data Availability Statement:** All relevant data are within the manuscript and its Supporting information files.

## Abstract

Regulation of the microtubule cytoskeleton is crucial for the development and maintenance of neuronal architecture, and recent studies have highlighted the significance of regulated RNA processing in the establishment and maintenance of neural circuits. In a genetic screen conducted using mechanosensory neurons of *C. elegans*, we identified a mutation in *muscleblind-1/mbl-1* as a suppressor of loss of kinesin-13 family microtubule destabilizing factor *klp-7*. Muscleblind-1(MBL-1) is an RNA-binding protein that regulates the splicing, localization, and stability of RNA. Our findings demonstrate that *mbl-1* is required cell-autonomously for axon growth and proper synapse positioning in the posterior lateral microtubule (PLM) neuron. Loss of *mbl-1* leads to increased microtubule dynamics and mixed orientation of microtubules in the anterior neurite of PLM. These defects are also accompanied by abnormal axonal transport of the synaptic protein RAB-3 and reduction of gentle touch sensation in *mbl-1* mutant. Our data also revealed that *mbl-1* is genetically epistatic to *mec-7* (β tubulin) and *mec-12* (α tubulin) in regulating axon growth. Furthermore, *mbl-1* is epistatic to *sad-1*, an ortholog of BRSK/Brain specific-serine/threonine kinase and a known regulator of synaptic machinery, for synapse formation at the correct location of the PLM neurite. Notably, the immunoprecipitation of MBL-1 resulted in the co-purification of *mec-7*, *mec-12*, and *sad-1* mRNAs, suggesting a direct interaction between MBL-1 and these transcripts. Additionally, *mbl-1* mutants exhibited reduced levels and stability of *mec-7* and *mec-12* transcripts. Our study establishes a previously unknown link between RNA-binding proteins and cytoskeletal machinery, highlighting their crucial roles in the development and maintenance of the nervous system.

## Author summary

Both RNA-binding proteins and microtubule cytoskeleton play important roles in the development and maintenance of a functional nervous system. A misregulation of mRNA

**Funding:** This work was supported by the National Brain Research Centre core fund from the Department of Biotechnology, The Wellcome Trust DBT India Alliance (IA/I/13/1/500874), and a grant from the Science and Engineering Research Board (SERB: CRG/2019/002194) to A.G.R. This work is also supported by a grant from the Science and Engineering Research Board (Grant # CRG/2018/00425) to S.B. The Caenorhabditis Genetics Centre is supported by the National Institutes of Health Office of Research Infrastructure Programs (P40 OD010440). The funders had no role in study design, data collection and analysis, decision to publish, or preparation of the manuscript.

**Competing interests:** The authors have declared that no competing interests exist.

processing and microtubule dynamics often contributes to neurodevelopmental and neurodegenerative disorders. Through a combination of genetic approaches and live cell imaging in the nematode *C. elegans*, we have uncovered a molecular connection between the RNA binding protein Muscleblind-1 (MBNL) and the microtubule cytoskeleton, specifically in neurons responsible for gentle touch sensation. Our investigations have revealed that Muscleblind-1/MBL-1 is involved in the optimal expression of the mRNAs that encode for the building blocks of microtubules, namely, the tubulin subunits MEC-7 (β-tubulin) and MEC-12 (α-tubulin) in touch neurons. Additionally, MBL-1 binds to another mRNA, *sad-1*, which encodes a kinase protein essential for synapse formation. In the absence of *mbl-1*, the microtubule cytoskeleton becomes unstable, leading to significant disruptions in the transport system within the neurite. Consequently, this impairment results in a reduction of sensory function performed by this set of neurons. The findings from our study provide a mechanistic insight into how RNA-binding proteins and the microtubule cytoskeleton are linked in neurons.

## Introduction

A highly ordered functional neuronal circuit comprises polarized nerve cells, which are compartmentalized into dendrites and axons for unidirectional reception and transmission of information. Several reports suggest that the structural and functional polarity of neurons relies on cytoskeletal elements within the neuron [1–4]. These cytoskeletal elements are regulated by intra- and extra-cellular signal transduction pathways during neuronal polarization [5–7]. The organization of the microtubule cytoskeleton in neurons plays a crucial role in directing neuronal polarization [8].

In vertebrate neurons, axonal microtubules are arranged in a parallel array with their plus-ends facing toward the synapse, while dendrites exhibit a mixed orientation of microtubules [8]. In invertebrate dendrites, the microtubule orientation is minus-end-out [9,10]. This polarized arrangement of microtubules is the basis for the proper localization of synaptic components and neurotransmitter release [8,11].

Recent reports have identified the critical roles of RNA-binding proteins in neuronal development [12] and synaptic transmission [13–15]. Disruptions in these genes can cause various neurological disorders [16–19]. The Muscleblind-like protein family (MBNL) is an evolutionarily conserved RNA-binding protein family that contains CCCH zinc-finger domains [20]. MBNL regulates alternative splicing, alternative polyadenylation, mRNA localization, RNA processing, and translation [21–23]. The role of MBNL in the neural pathogenesis of myotonic dystrophy type 1(DM1), has been extensively discussed [24–27].

In the mouse brain, loss of MBNL leads to misregulated alternative splicing and polyadenylation, resulting in defects in motivation and spatial learning, and abnormal REM (Rapid Eye Movement) sleep [21,28–31]. The MBNL family in mammals consists of three proteins: MBNL1, MBNL2, and MBNL3, encoded by three different genes, each with several isoforms [32]. The functions of these different isoforms of MBNL vary, corresponding to their differential localization [33–36]. A recent report in *Drosophila* showed that Muscleblind (MBL) is expressed in the nervous system and regulates alternative splicing of *Dscam2* during the development of the nervous system [37]. In *C. elegans*, the Muscleblind homolog MBL-1 binds to multiple transcripts and regulates life span [38]. MBL-1 has also been shown to regulate synapse formation in motor neurons [39]. In touch neurons, it regulates the splicing of the *sad-1* (SAD/BRSK kinase) gene [40]. Although there is an indication of a functional link between

Muscleblind and microtubule cytoskeleton [41], a comprehensive understanding of how Muscleblind regulates the microtubule cytoskeleton in neurons remains unclear.

*C. elegans* neurons have been extensively used to study the mechanisms of neuronal polarization *in vivo* [42–46]. Specifically, studies using mechanosensory neurons responsible for gentle touch sensation, have provided insights into the role of the microtubule cytoskeleton in neuronal development and maintenance [7,47–49]. In this study, we identified a mutation in the *muscleblind-1* gene, as a suppressor of the loss of kinesin-13 family microtubule depolymerase *klp-7*, which affects the neuronal phenotypes observed in the ALM and PLM touch neurons. We found that Muscleblind-1 (MBL-1) is required for axonal growth and synapse formation in the PLM neurons. Using live imaging of plus-end binding protein (EBP-2::GFP), and synaptic protein (GFP::RAB-3), we observed compromised microtubule stability in the absence of *mbl-1*, leading to reduced vesicular transport. We found that *mbl-1* is epistatic to both *mec-7* and *mec-12* for controlling axon outgrowth in PLM neurons. We also demonstrated that MBL-1 binds and regulates the stability of transcripts encoding these touch neuron-specific tubulins. Moreover, we found that *mbl-1* is epistatic to *sad-1*, which regulates the synaptic organization in *C. elegans* neurons [50]. Our data illustrate that *mbl-1* regulates *sad-1* in controlling synapse positioning in PLM neurons. Collectively, our findings suggest that MBL-1 regulates the cytoskeletal machinery in neurons by regulating mRNA stability.

## Results

### Loss of *muscleblind-1 (mbl-1)* suppresses the ectopic neurite growth phenotype in touch neurons of *klp-7(0)* mutant

In *C. elegans*, six mechanosensory neurons are responsible for gentle touch sensation, including a pair of Anterior Lateral Microtubule (ALM) and a pair of Posterior Lateral Microtubule (PLM) neurons (white arrowheads, Fig 1A). ALM and PLM neurons extend their axons laterally towards the anterior side and establish connections with their respective postsynaptic neurons through a ventral synaptic branch (white arrows, Fig 1A). Additionally, the PLM neuron extends a short posterior neurite into the tail of the animal (double-sided white arrow, Fig 1A). Previous research has shown that loss of microtubule depolymerizing protein kinesin-13/ KLP-7 leads to an ectopic extension of the posterior neurite from the ALM cell body (yellow arrow, Fig 1A) due to increased stabilization of the microtubule cytoskeleton [7]. Similarly, an overextension of the posterior neurite is observed in the PLM neuron (double-sided white arrow, Fig 1A). This neurite overgrowth phenotype in the *klp-7* mutant can be suppressed by destabilizing microtubules using colchicine or by the loss of tubulin subunits [7]. Based on this, we hypothesized that conducting an EMS mutagenesis suppressor screen in the *klp-7* mutant background could help identify genes and associated pathways involved in regulating the microtubule cytoskeleton in neurons.

In this suppressor screen, one of the mutants isolated, named *ju1128*, showed suppression of the ectopic extension phenotype in the ALM neurons in *klp-7(0)* background (red arrow, Fig 1A and 1C). Further analysis indicated that *ju1128* mapped to the locus of the *mbl-1* gene, which codes for the RNA binding protein Muscleblind-1/MBL-1. Several lines of evidence suggest that *ju1128* is an allele of the *mbl-1* gene. Firstly, the SNP chromosomal mapping indicated a linkage between the suppression and the right arm of the X chromosome (red arrowhead, S1A Fig). The whole-genome sequencing and Cloudmap analysis of the outcrossed suppressor [51] allowed us to narrow down the mutation to the *mbl-1* gene locus on the X-chromosome (S1B Fig). By annotating the SNPs obtained from the Cloudmap analysis [51], we identified a C-T transition at the 17002646[th] base pair position of chromosome X (Fig 1B). This specific transition occurs at the 85[th] nucleotide in the third exon of the *mbl-1* gene, resulting in the

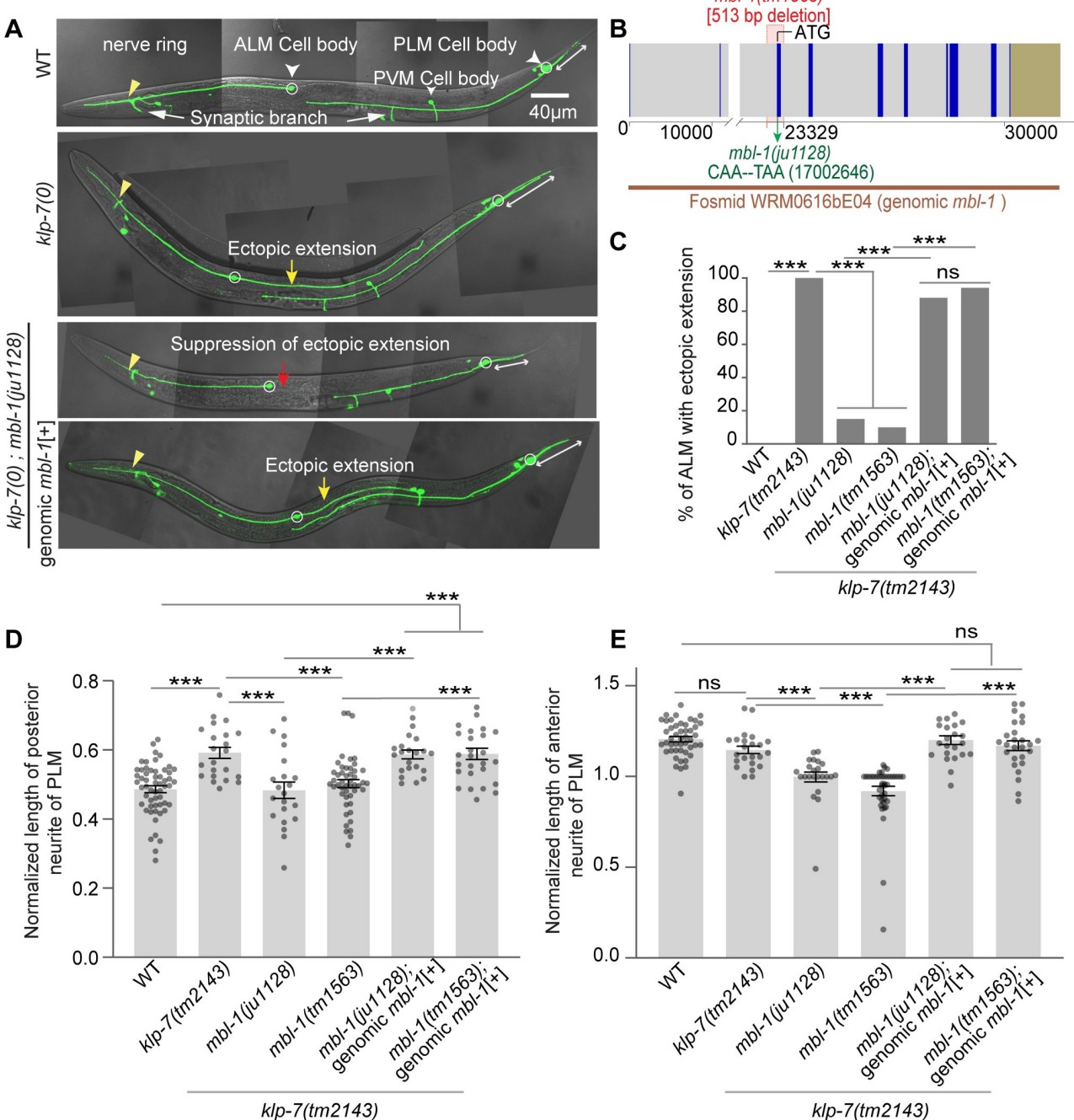

**Fig 1. Characterization and mapping of *ju1128* mutation.** (A) Confocal images of ALM and PLM neurons of a wild-type, a *klp-7(0)*, a suppressor *klp-7(0); mbl-1(ju1128)* and a *klp-7(0); mbl-1(ju1128); genomic mbl-1[+]* larval-stage four (L4) animal, expressing *muIs32 (Pmec-7*::GFP*)* reporter. The ectopic extension of the ALM posterior process in *klp-7(0)* is marked by a yellow arrow while PLM posterior process is marked by a double-sided white arrow. The suppression of *klp-7(0)* ectopic extension in suppressor *klp-7(0); ju1128* is marked by the red arrow. (B) The schematic of the exon and intron of the *mbl-1* gene and the genomic positions of *ju1128* and *tm1563* alleles in the *mbl-1* locus. The fosmid WRM0616bE04, which exclusively covers the *mbl-1* gene, is also shown. (C) Quantification of percentage of ALM neurons with ectopic extension in *klp-7(0); mbl-1(ju1128)*, *klp-7(0); mbl-1(tm1563)*, *klp-7(0); mbl-1(ju1128); genomic mbl-1[+]* and *klp-7(0); mbl-1(tm1563); genomic mbl-1[+]* backgrounds, where genomic *mbl-1* is the *mbl-1* fosmid WRM0616bE04. N = 3–5 independent replicates, n (number of neurons) = 100–150. (D-E) The normalized length of posterior (D) and anterior neurite (E) of PLM, in *klp-7(0)*, suppressors *klp-7(0); mbl-1(ju1128)* and *klp-7(0); mbl-1(tm1563)* and the rescue backgrounds. Normalized length = (Actual length/distance between the PLM cell body and vulva for the anterior neurite, and the distance between the PLM cell body to the tip of the tail for the posterior neurite). N = 3–4 independent replicates, n (number of neurons) = 21–47. For C, ***P<0.001; Fisher's exact test. For D-E, ***P<0.001; ANOVA with Tukey's multiple comparison test. Error bars represent SEM, ns, not significant.

introduction of a premature stop codon instead of the expected glutamine amino acid in
*ju1128*.

To confirm the mapping results, we expressed the fosmid WRM0616bE04, which exclusively contains the complete *mbl-1* gene locus, in the *klp-7(0); mbl-1(ju1228)* background. This fosmid, referred to as genomic *mbl-1*, rescued the suppression of the ectopic extension phenotype in the ALM neurons (yellow arrow, Fig 1A and 1C) as well as the overgrowth phenotype of the posterior neurite in the PLM neurons (double-sided white arrow, Fig 1A and 1D). Another allele of *mbl-1*, *tm1563*, which is a deletion mutation of the third exon (Fig 1B), also suppressed the ectopic neurite growth phenotypes in *klp-7(0)* (Fig 1C–1E). Both the *ju1128* and *tm1563* mutations in *mbl-1* resulted in stunted growth of the anterior neurite of PLM in *klp-7(0)* background, and this phenotype was rescued by the genomic fragment of *mbl-1* (Fig 1A and 1E). These findings collectively confirmed that *ju1128* is a mutation in the *mbl-1* gene and that loss of function of *mbl-1* suppresses the neuronal phenotypes observed in the *klp-7(0)*.

## Muscleblind-1(MBL-1) regulates axon growth and synapse formation in PLM neuron

To investigate the role of the *mbl-1* gene in touch neuron development, we conducted experiments by removing the *klp-7(tm2143)* allele from the suppressor background. In the wild-type background, the PLM anterior neurite extends anteriorly and crosses the vulva, frequently approaching the ALM cell body (yellow arrowhead, Fig 2A). Additionally, it forms a ventral synaptic branch near the vulva (white arrowhead, Fig 2A). However, in both mutant alleles of *mbl-1*, the anterior neurite either terminates (yellow arrowhead is the termination point) before reaching the vulva or at the vulval position (marked by an asterisk, Fig 2A), resulting in a 'short neurite' phenotype. Approximately 88% of the PLM neurites in *mbl-1* mutants exhibited this short neurite phenotype (Fig 2A and 2B). Since the Muscleblind-1 protein has known roles in both muscles [20] and neurons [37,39], we wanted to investigate the tissue-specific function of *mbl-1* in neurite development. When *mbl-1* cDNA (C isoform) was transgenically expressed using the muscle-specific promoter *Pmyo-3* in *mbl-1(0)* mutants, there was no rescue of the 'short neurite' phenotype (Fig 2B). However, pan-neuronal expression of *mbl-1* cDNA (C isoform) using *Prgef-1::mbl-1*, as well as mechanosensory neuron-specific expression using P*mec-4::mbl-1*, in the *mbl-1(0)* background, resulted in significant rescue of the short neurite phenotype, comparable to the rescue obtained using the *mbl-1* genomic DNA (Fig 2A and 2B). This suggests that *mbl-1* functions in a cell-autonomous manner to regulate neurite length in PLM neurons.

It is known that the PLM axon extends anteriorly and establishes physical contact with the BDU interneuron through a gap junction synapse at its distal end [52]. In *mbl-1(0)* mutants due to the short neurite phenotype, the physical contact between the PLM axon and the BDU interneuron is lost (S2A–S2C Fig). By using the gap junction reporter UNC-9::GFP [52] in the *mbl-1(0)* mutants, we observed the absence of gap junction at the tip of the PLM anterior neurite (S2D–S2F Fig).

In the wild-type background, the axon of the ALM touch receptor neuron forms a synapse at the nerve ring (white arrowhead Fig 2A) and extends anteriorly, terminating near the tip of the head (yellow arrowhead, Fig 2A). Similar to PLM in *mbl-1(0)* mutants, the ALM neuron also exhibits a short neurite phenotype that terminates at the nerve ring (yellow arrowhead, Fig 2A).

The short neurite phenotype observed in the PLM neurons of *mbl-1(0)* mutants is not caused by the degeneration of neurons, as no neurite fragmentation was observed until the

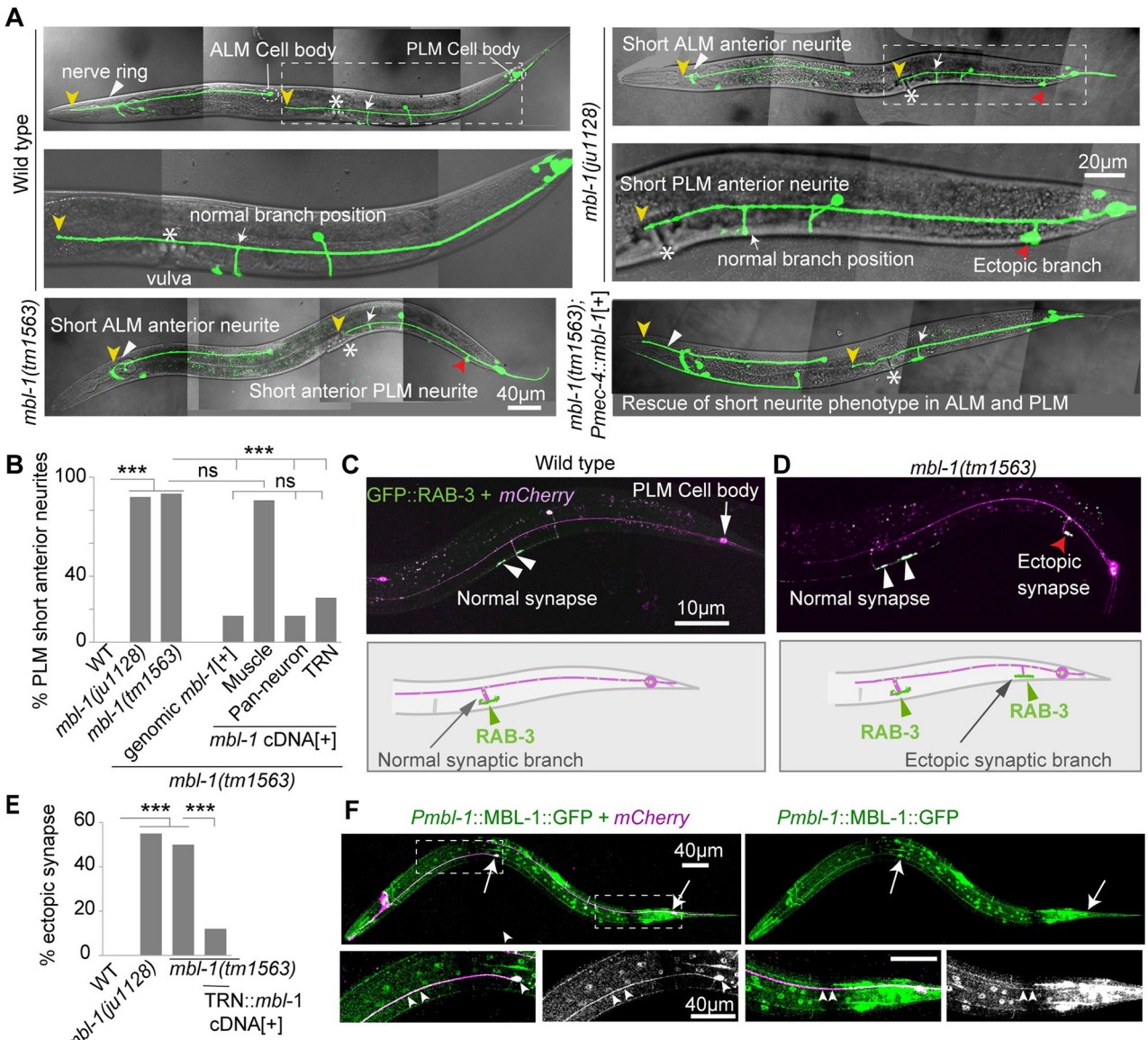

**Fig 2. *mbl-1* mutants display defects in axon growth, and synapse formation in PLM neurons.** (A) Confocal images of ALM and PLM neurons at the L4 larval stage in WT, *mbl-1(tm1563)*, *mbl-1(ju1128)*, and *mbl-1(tm1563); Pmec-4::mbl-1*(C-isoform)[+] background. The ends of ALM and PLM anterior neurites are marked with a yellow arrowhead, and the vulval location is marked with an asterisk (*). A white arrow marks the normal synapse position in *mbl-1(0)*. The presence of ectopic synapse in *mbl-1(tm1563)* is marked by a red arrowhead. (B) Quantification of the percentage of short PLM anterior neurites at the L4 stage in two alleles of *mbl-1(0)* and different rescue backgrounds. N = 4–5 independent replicates, n (number of neurons) = 100–200. (C and D) Image and schematics of synapse-position defect in PLM in both wild-type (C) and *mbl-1(tm1563)* (D) at the L4 stage. PLM synapse (white arrowheads) was visualized with the *Pmec-7*::GFP::RAB-3 (*jsIs821*) transgene and *Pmec-4::mCherry* (*tbIs222*) was used for visualizing PLM neurons. The presence of an ectopic synapse in the *mbl-1(tm1563)* background is marked by a red arrowhead. (E) Quantification of the percentage of neurites with an ectopic synapse in *mbl-1(ju1128)*, *mbl-1(tm1563)*, and *mbl-1(tm1563); Pmec-4::mbl-1*[+] backgrounds. N = 3–4 independent replicates, n (number of neurons) = 80–100. (F) Confocal images of ALM and PLM neurons (white arrows) in animals expressing *Pmbl-1*::MBL-1::GFP (*wgIs664*) in P*mec-4:mCherry* (*tbIs222*) background. The enlarged versions of the anterior neurites were shown below with arrowheads indicating the enrichment of MBL-1::GFP in PLM and ALM neurons. Statistical comparisons were made using the χ2 (Fisher's exact) test for B and E, ***P <0.001. Fisher's exact test. ns, not significant.

day-6 adult stage (S2I Fig). However, we did observe 33% degeneration of PLM neurites in the day-7 adult stage, which was comparable to the wild-type (S2G–S2I Fig).

In addition to the short neurite phenotype, an ectopic branch near the PLM cell body (red arrowhead, Fig 2A) was also observed in *mbl-1(0)* mutants. We used a presynaptic marker (*Pmec-7*::GFP::RAB-3) [53] to characterize this ectopic branch. In the wild-type background, GFP::RAB-3 is enriched along the ventral cord at the end of the ventral branch of PLM near the vulva (white arrowhead, Fig 2C). However, in *mbl-1(0)* mutants, GFP::RAB-3 is also enriched at the tip of the ectopic synapse near the PLM cell body (red arrowhead, Fig 2D), in addition to the original synapse (white arrowhead, Fig 2D). Approximately 50% of PLM neurons in both mutant alleles of *mbl-1* exhibited these ectopic synapses (Fig 2E), and this phenotype was rescued by extrachromosomal expression of *mbl-1* cDNA in mechanosensory neurons (Fig 2E). To further confirm this result, we used another presynaptic active zone marker ELKS-1 (ELKS-1::TagRFP) [54], and it showed similar localization in these ectopic synapses near the PLM cell body as observed with GFP::RAB-3 (red arrowhead, S3A–S3C Fig). These observations suggest that MBL-1 regulates synapse formation and positioning in PLM neurons.

To determine if the mutation in the *mbl-1* gene affects other classes of neurons, we visualized the D-type GABAergic motor neurons using the *Punc-25*::GFP reporter transgene. In the *mbl-1(0)* mutant, we noticed gaps on the dorsal side, (red arrowhead, S3D–S3F Fig) indicating a possible defect in neurite growth, similar to what was observed in the DA9 neuron of *mbl-1* mutants [39]. This suggested that the role of *mbl-1* in regulating neurite growth is not specific to TRN, rather more general to different classes of neurons.

To examine the localization pattern of MBL-1, we used an MBL-1::GFP translational reporter [55] under its native promoter and observed GFP expression in several tissues throughout the body of the animal (Fig 2F). MBL-1::GFP was highly enriched in ALM and PLM neurons (Fig 2F), with localization detected in the cell bodies (White arrows, Fig 2F) as well as in the PLM and ALM neurites (white arrowheads, Fig 2F).

Several RNA-binding proteins have been identified in the nervous system of *C. elegans*, including MEC-8/RBMPS (RNA-binding protein, mRNA processing factor), MSI-1/Musashi (enables RNA-binding activity), UNC-75/CELF5 (CUGBP Elav-like family protein, enables ssRNA-binding activity), and EXC-7/ELAVL4 (enables ssRNA-binding activity) [56]; we investigated if any of them played a role in regulating neurite length or synapse formation in TRN. However, no noticeable morphological defects were observed in loss of function mutants of these genes (S3G and S3H Fig). These observations suggest that the RNA-binding protein, MBL-1 is specifically required for neurite growth and synapse formation in touch neurons.

## MBL-1 regulates microtubule polarity and stability in the anterior neurite of PLM neurons

Since the *mbl-1* mutant suppressed the neuronal overgrowth phenotype caused by the increased stabilization of microtubules in the *klp-7* mutant, we looked for any possible defects in the microtubule cytoskeleton in *mbl-1(0)*. We conducted time-lapse live imaging using the EBP-2::GFP reporter, which binds to the plus ends of growing microtubules, to examine the microtubule dynamics [57]. By analyzing kymographs generated from regions of interest in the anterior and posterior neurites of PLM neurons (Fig 3A), we determined the polarity of the microtubules from the direction of growth of microtubule plus ends (Fig 3B). In the wild-type background, the majority of the EBP-2::GFP movements in the PLM anterior neurite were directed away from the cell body (plus-end-out, green trace), whereas the posterior neurite showed movements in both directions, away from and towards the cell body (minus-end-

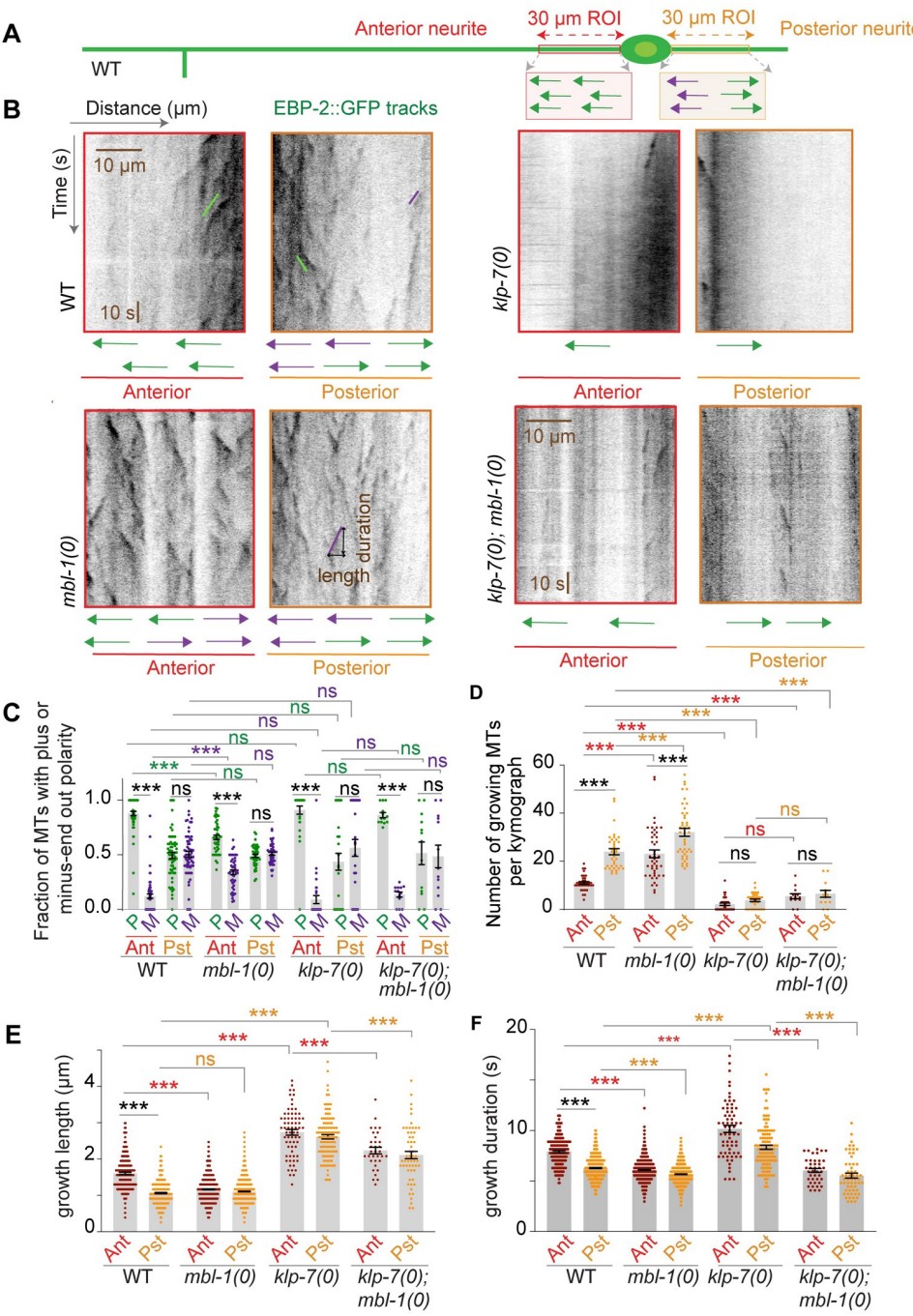

**Fig 3. *mbl-1* mutant affects microtubule dynamics in PLM neuron.** (A) Schematic of the PLM neuron showing the regions of interest (ROIs), marked in red and orange for anterior and posterior processes, respectively. These ROIs were used for the analysis of time-lapse movies of *Pmec-4*::EBP-2::GFP (*juIs338*). (B) Representative kymographs of EBP-2::GFP reporter obtained from the above-mentioned ROIs in wild type, *mbl-1(0), klp-7(0) and klp-7(0); mbl-1(0)* backgrounds. The green and purple traces on kymographs represent microtubule growth events away from the cell body (Plus end out) and towards the cell body (Minus end out), respectively. (C) The histogram is showing the fraction of microtubules with 'plus-end-out' (P) or 'minus-end-out' (M) polarity in the anterior and posterior processes in the mentioned genetic backgrounds. N = 3–5 independent replicates, n (number of worms) = 44–62. (D) The histogram represents the number of diagonal tracks (growing microtubules) in PLM anterior and posterior processes in the mentioned genetic backgrounds. N = 3–5 independent replicates, n (number of worms) = 36–46. (E and F) Growth length (E) and growth duration (F) of the microtubule growth events (tracks), measured from net pixel shift in the X and Y axis respectively, from kymographs shown in B. N = 3–5 independent replicates, n (number of tracks) = 227–1074. For C-F, ***P <0.001; ANOVA with Tukey's multiple comparison test. Error bars represent SEM. ns, not significant.

out, purple trace, Fig 3A and 3B), as we have reported previously [7]. Plotting the fraction of microtubule tracks with 'plus-end-out' or 'minus-end-out' orientation (Fig 3C) revealed a significant decrease in the percentage of microtubules with plus-end-out orientation in the PLM anterior neurite of *mbl-1(0)* animals compared to that of wild-type animals (Fig 3C). The microtubule polarity in the posterior neurite of *mbl-1(0)* was similar to the wild-type (Fig 3C). Additionally, we observed an increased number of EBP-2::GFP tracks in both the anterior and posterior neurites of PLM in *mbl-1(0)* (Fig 3D) compared to the wild-type. However, the growth length and duration of these tracks were significantly shorter in the PLM anterior neurite of *mbl-1(0)* compared to the wild-type, while in the PLM posterior neurite only the growth duration was shorter than the wild-type (Fig 3E and 3F). These observations suggest that *mbl-1(0)* exhibits increased microtubule dynamics and a defect in the microtubule polarity in the anterior neurite.

Furthermore, we investigated whether the *mbl-1* mutation alters the microtubule dynamics phenotype in the *klp-7(0)* background. In the *klp-7(0)* mutant, there was a reduction in the number of EBP-2 tracks, and their growth length and duration were increased (Fig 3B–3F), as we have previously reported [7]. Interestingly, the *mbl-1(0)* mutation in the *klp-7(0)* background did not change the number of EBP-2 tracks or the distribution of plus-end-out or minus-end-out tracks (Fig 3C and 3D). However, the growth duration and length of the tracks were significantly reduced in *klp-7(0); mbl-1(0)* double mutant animals as compared to *klp-7(0)* alone (Fig 3E and 3F). This indicates that the *mbl-1(0)* mutation partially suppresses the increased stability phenotype of microtubules observed in the *klp-7(0)* background.

In summary, these results provide evidence that MBL-1 regulates microtubule polarity and stability in PLM neurons.

## MBL-1 regulates axonal transport and gentle touch sensation in PLM neurons

Since the loss of *mbl-1* affects microtubule orientation and stability, we examined axonal transport in *mbl-1(0)* mutants using a GFP reporter for the presynaptic protein RAB-3 [53]. We imaged GFP::RAB-3 in similar ROIs (Fig 4A) as those used for imaging EBP-2::GFP. In the *mbl-1(0)* mutants, the number of anterograde and retrograde transport events was reduced in the PLM anterior neurite compared to the wild type (Fig 4B and 4C). However, in the PLM posterior neurite, the number of transport events was similar to that in the wild type (Fig 4B and 4D). We also observed that in the PLM anterior neurite in *mbl-1(0)*, most of the particles were static (Fig 4B). The run length of the transport events was shorter in the *mbl-1(0)* compared to the wild type in both the anterior and posterior PLM neurites (Fig 4E and 4F).

Considering the significant impact on the transport of synaptic proteins, we wondered whether the function of this neuron is compromised in the *mbl-1* mutant. To address this, we assessed the gentle touch response in the *mbl-1(0)* mutant. The anterior and posterior touch response was significantly reduced in both *mbl-1* mutant alleles, and this reduction was rescued by introducing a genomic copy of the *mbl-1* transgene (Fig 4G and 4H).

Collectively, our findings suggest that loss of MBL-1 compromises the microtubule cytoskeleton in the PLM axon, which consequently disrupts the axonal transport and impairs the function of this neuron.

## *mbl-1* interacts with the genes that regulate the microtubule cytoskeleton to control neurite growth of PLM neurons

MBL-1 is a Zinc finger family RNA-binding protein that preferentially binds to the CGCU sequence of target RNA [58] (S4A Fig). To understand how MBL-1 regulates MT dynamics

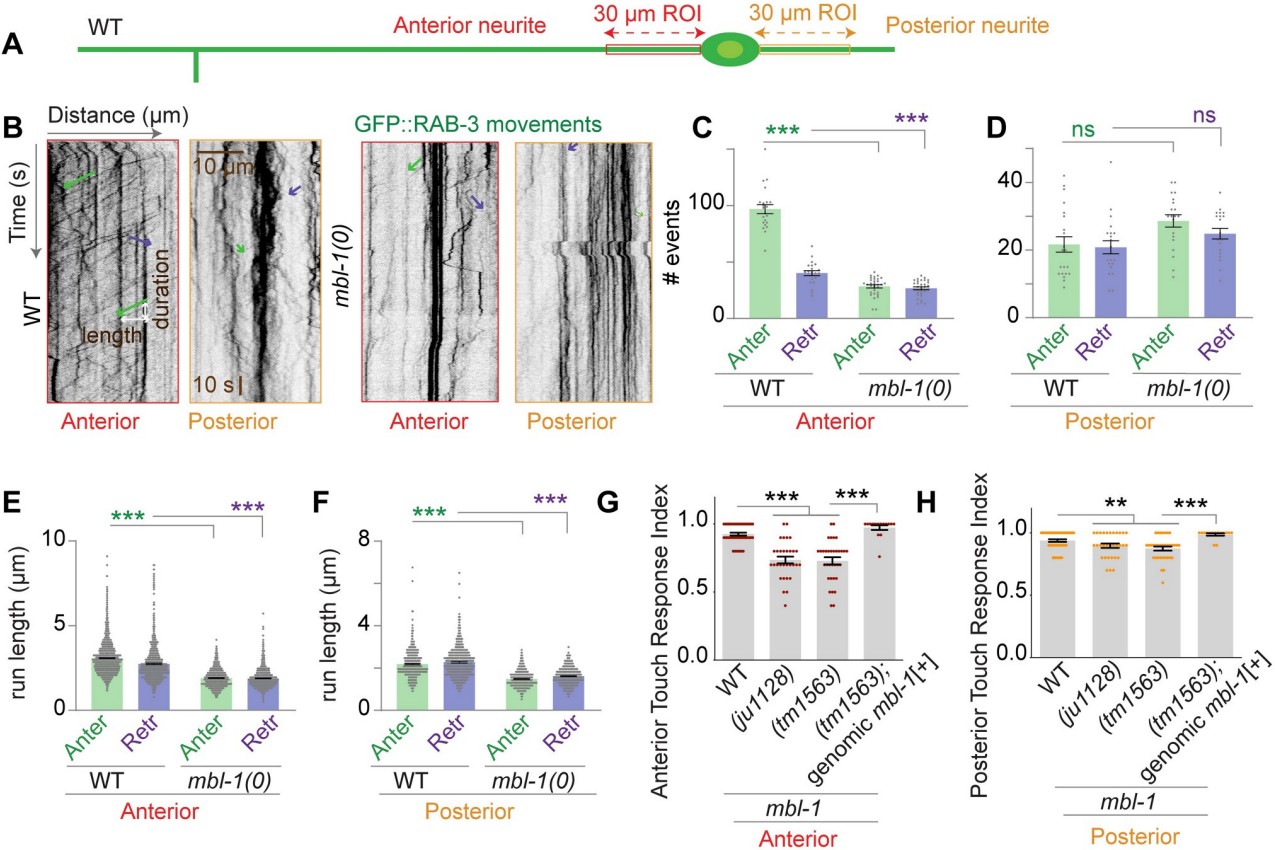

**Fig 4. *mbl-1* mutant displays defect in axonal transport and gentle touch sensation.** (A) Schematic of the PLM neuron showing the regions of interest (ROIs), marked in red and orange for anterior and posterior processes, respectively. These ROIs were used for the analysis of time-lapse movies of *Pmec-7*::GFP::RAB-3 (*juIs821*). (B) Representative kymographs of time-lapse movies of *Pmec-7*::GFP::RAB-3 (*juIs821*) as obtained from the above-mentioned ROIs (A) in wild type, and *mbl-1(0)* background. The green and purple traces on kymographs represent movement events away from the cell body (anterograde) and towards the cell body (retrograde) respectively. (C and D) Quantification of the anterograde (Anter) and retrograde (Retr) movement events of GFP::RAB-3 particles obtained from kymographs shown in B in PLM anterior (C) and posterior (D) processes in wild-type and *mbl-1(0)*, N = 3–5 independent replicates, n (number of worms) = 21–32. (E and F) run length of GFP::RAB-3 movements in PLM anterior (E) and posterior (F) processes, measured from net pixel shift in X-axis direction as shown in the kymograph, N = 3–5 independent replicates, n (number of tracks) = 456–2131. (G and H) The histogram shows the anterior (G) and posterior (H) gentle touch response index of the worm in the wild-type, two alleles of the *mbl-1* gene, and rescue background. N = 3 independent replicates, n (number of worms) = 31–50. For C-H, ** P < 0.01; ***P <0.001; ANOVA with Tukey's multiple comparison test. Error bars represent SEM. ns, not significant.

and axon growth in PLM neurons, as well as to identify potential interacting partners, we screened for transcripts containing an MBL-1 binding site using the oRNAment database [59]. We focused on mRNAs expressed in the PLM neuron, as reported in the CeNGEN database [56], that resulted in the identification of 2,000 MBL-1 targets (Fig 5A) (S1 Table). Using gene ontology (GO) analysis, we short-listed four sets of genes from this pool for further analysis, categorized based on their involvement in (1) microtubule-based processes, (2) axon development, (3) regulation of synapse structure, and (4) axodendritic transport (S2 Table) (Fig 5A). In this study, for each category we analyzed a subset of genes for genetic interactions with *mbl-1*, based on their known involvement in neuron development (S3 Table).

In the microtubule-based processes category, we identified 129 genes (S2 Table) (Fig 5A), of which we tested 21 (S3 Table) that are either part of the microtubule structure or regulators of microtubule dynamics. For example, we examined genes such as *mec-7 and mec-12*, which are touch neuron-specific tubulins and crucial for microtubule assembly, as well as CRMP-2

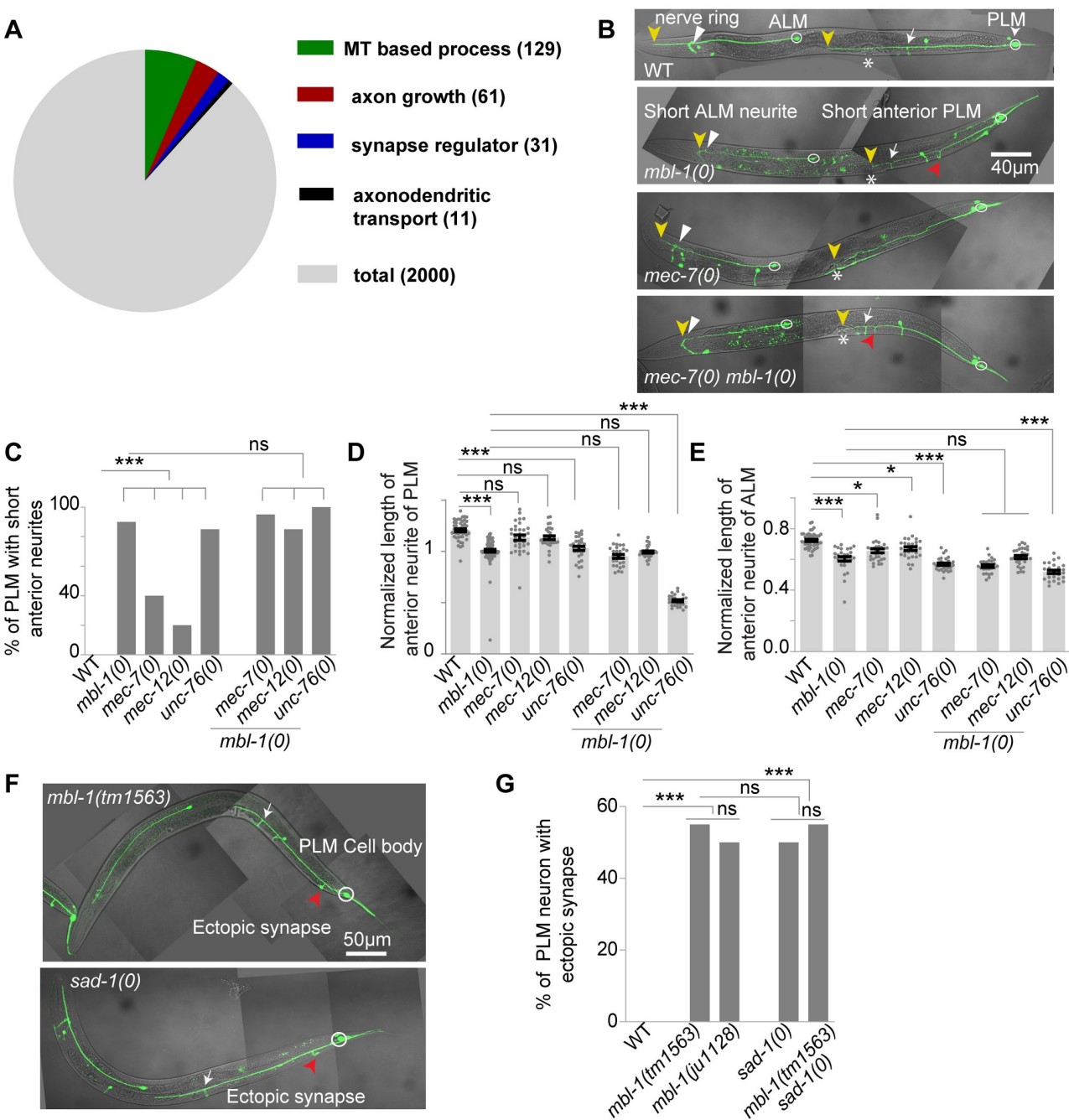

**Fig 5. *mbl-1* genetically interacts with mutants affecting microtubule cytoskeleton and SAD-1 kinase.** (A) Pie chart showing the result of Gene ontology (GO) analysis of putative targets (with MBL-1 binding site) of MBL-1 that are expressed in PLM neurons (B) Representative confocal images of ALM/PLM neurons in wild-type, *mbl-1(0)*, *mec-7(0)* and *mbl-1(0) mec-7(0)* L4 stage worms, expressing *muIs32 (Pmec-7*::GFP*)* reporter. The yellow arrowheads indicate the ends of anterior neurites in PLM/ALM neurons. The vulval position is marked by an asterisk (*). The normal synapse position is marked by a white arrow. (C) The histogram is showing the percentage of PLM neurons with a short anterior neurite. (D) The normalized length of the anterior neurite of PLM neuron in different backgrounds. For PLM neurons, Normalized length = (Actual length of neurite/distance between the PLM cell body and vulva) (E) Quantification of the normalized length of the anterior neurite of ALM neurons. For ALM neuron, normalized length = (Actual length of neurite /distance between the vulva and the tip of the nose). (F-G) The representative confocal images (F) and quantification (G) of ectopic synapses (marked with red arrowhead) in the *mbl-1(0)* and *sad-1(0)* L4 stage worms, expressing *muIs32 (Pmec-7*::GFP*)* reporter. For C-E, and G independent replicates (N) = 3–5 and the number of neurons (n) = 30–150. For C and G, ***$P<0.001$; Fisher's exact test. For D-E, *$P < 0.05$; ***$P <0.001$; ANOVA with Tukey's multiple comparison test. Error bars represent SEM, ns, not significant.

(*unc-33*) which helps in microtubule polymerization [47,60]. In the axon growth-related genes category, we found 61 genes (S2 Table) (Fig 5A), and we tested 16 (S3 Table) known to play important roles in axon growth, including *unc-51* and *unc-53* [61–63].

For the regulators of the synapse structure category, we identified 31 genes (S2 Table) (Fig 5A) and examined 4 (S3 Table) known to regulate synapse development, such as *sad-1*/SAD BRSK kinase [5,6].

In the axodendritic transport-related genes category, we found 11 genes (S2 Table) (Fig 5A) and tested 3 (S3 Table) known to be regulators of motor-based transport, such as *unc-104* [64,65].

Additionally, a recent report identified a set of 235 genes that are downregulated in the *mbl-1* mutant [38]. We further analyzed this set, using GO analysis, to identify candidates within the aforementioned categories. This analysis led us to *mec-7*, *mec-12*, and *klp-13*, which are directly involved in microtubule-based processes. However, we did not find any candidates linked to axon development, synapse structure, or axodendritic transport.

We did a phenotypic assessment of touch receptor neurons in the mutants of the above-mentioned candidate genes, for short neurite or ectopic synapse phenotype in the PLM neuron, as observed in the *mbl-1(0)* mutant.

We observed that loss of function mutations in TRN-specific tubulins, *mec-7*, and *mec-12* resulted in a short neurite phenotype in the PLM anterior neurite (Fig 5B–5D). In the *mec-7* mutant, the anterior neurite of PLM terminates near the vulva (marked by an asterisk) resembling the phenotype seen in the *mbl-1* mutant alone. We also found that mutation in the vesicular adaptor protein (*unc-76)* leads to a short neurite phenotype in the PLM anterior neurite (red arrow, Fig 5B–5D), similar to *mbl-1(0)*. The transcript of each of these genes has putative MBL-1/MBNL-1 binding sites with varying binding probabilities, as inferred from *in-silico* data (S4A–S4D Fig). Further analysis in the *mbl-1(0) mec-7(0)* double mutant revealed that the length of the PLM anterior neurite remained the same as that observed in the *mbl-1(0)* single mutant, indicating that the phenotype is not additive (Fig 5B–5D). Similarly, the short neurite phenotype in *mec-12(0); mbl-1(0)* double mutant was not enhanced compared to the *mbl-1* single mutant (Fig 5B–5D). However, in the *mbl-1(0); unc-76(0)* double mutant, the length of the PLM axon was significantly shorter than in either single mutant (Fig 5D). A similar observation was made in the ALM neurons (yellow arrowhead, Fig 5B and 5E).

As observed in *mbl-1(0)* mutant, and as it is already known for tubulin mutants, they exhibit defects in both anterior and posterior gentle touch responses. We examined how this behavior was affected in the double mutants. We did not observe any additive effects in the anterior or posterior touch response index in *mec-7(0) mbl-1(0)* or *mec-12(0); mbl-1(0)* double mutants (S4E and S4F Fig). These results suggest that since the *mbl-1* and tubulin genes have a similar phenotype and are epistatic in regulating the neurite length, they are genetically functioning in the same pathway. Given that *mbl-1* and *unc-76* mutations have an additive effect, these two might be working through two separate pathways to regulate neurite length. However, we did not observe any synaptic defect in these mutants.

A recent study demonstrated that MBL-1 regulates the splicing of *sad-1* in ALM neurons, and *in-silico* analysis revealed putative binding sites in *sad-1* [40] (S4D). We found that the *sad-1* mutants exhibited a synapse defect similar to that of the *mbl-1* mutant animals (red arrowheads, Fig 5F and 5G). Additionally, in the *sad-1(0) mbl-1(0)* double mutants, the extent of the ectopic synapse defect was the same as in single mutants (Fig 5G). These results suggest that *mbl-1* and *sad-1* are genetically functioning in the same pathway for synapse positioning in PLM neurons.

Taken together these results indicate that *mbl-1* collaborates with *mec-7* and *mec-12* in the same genetic pathway to regulate neurite length. Furthermore, it genetically interacts with *sad-1* to control synapse formation in PLM neurons.

## MBL-1 regulates the expression of *mec-7* and *mec-12* mRNAs

MBL-1 protein has well-documented functions in alternative splicing, RNA stability, and RNA localization [21–23]. As discussed, MBL-1 functions in conjunction with *mec-7*, *mec-12*, and *sad-1* to regulate TRN development. To understand how MBL-1 regulates these transcripts, we conducted quantitative RT-PCR analysis of *mec-7*, *mec-12*, *sad-1*, and *unc-76* in wild-type and *mbl-1(0)* animals. We observed no changes in the lengths of the RT-PCR products of *mec-7* and *mec-12* genes in *mbl-1(0)*, a result verified using multiple primers, suggesting that MBL-1 most likely does not regulate the splicing of these transcripts (Fig 6A and 6B, S5A and S5B Fig). However, we did observe a decrease in the transcript levels of *mec-7* and *mec-12* in the *mbl-1(0)* mutant animals (Fig 6A and 6B). This reduction was not observed in control genes such as *aak-2* or *tba-1*, nor putative targets *sad-1* or *unc-76* (Fig 6A and 6B, S5B Fig). Quantitative RT-PCR analysis further confirmed a significant decrease in total *mec-7* and *mec-12* levels in the *mbl-1(0)* background compared to the control (Fig 6B and 6C). Another set of primers yielded similar results (S5A and S5C Fig), supporting the conclusion that MBL-1 regulates the transcript levels of *mec-7* and *mec-12* genes.

However, no changes were observed in the transcript length or the total amount of *sad-1* in the *mbl-1(0)* background (Fig 6C and S5B and S5C Fig). Similarly, no changes were observed in the transcript length or transcript levels of *unc-76* in *mbl-1(0)* animals (S5B and S5C Fig). As we have already seen (Fig 5B–5E), *unc-76* works genetically in a parallel pathway with *mbl-1* and is likely regulated by a separate mechanism.

Next, we investigated whether MBL-1 regulates the transcription or stability of its target mRNAs. To test this, we fed the animals Actinomycin D (Act D), a well-known transcription inhibitor across various model organisms including *C. elegans* [66–69]. By blocking new transcript production, we could assess changes in pre-existing *mec-7* and *mec-12* transcript levels in *mbl-1(0)*. In wild-type animals, Act D treatment resulted in a 25% reduction in *mec-7* transcript levels ($0.7462 \pm 0.0437$ fold difference relative to untreated, $p < 0.05$) and an 18% reduction in *mec-12* transcript levels ($0.8245 \pm 0.0503$ fold difference relative to untreated) (Fig 6D and 6E), confirming the effectiveness of Act D in blocking transcription.

Comparing *mec-7* and *mec-12* transcripts in ActD-treated wild-type and *mbl-1(0)* mutant animals we observed additional reductions in the transcript levels in *mbl-1(0)*. In ActD-treated *mbl-1(0)* animals *mec-7* transcript levels decreased by 58% (0.4170 fold difference relative to untreated) compared to a 25% reduction in treated wild-type animals. Similarly, *mec-12* transcript levels in ActD-treated *mbl-1(0)* animals showed a 33% reduction (0.6689 fold difference relative to untreated), compared to an 18% reduction in treated wild-type animals (Fig 6D and 6E). This additional reduction suggests increased instability of the pre-existing *mec-7* and *mec-12* transcripts in the absence of MBL-1 and new transcription. Notably, the ActD-dependent reduction in *mec-7/mec-12* transcript levels in the wild-type background indicates inherent turnover of these transcripts. If MBL-1 did not influence their stability, degradation patterns would have been similar in both wild-type and *mbl-1(0)* mutants. The observed additional decrease during transcriptional blockade may be attributed to increased instability of the pre-existing transcripts in the absence of MBL-1.

To validate the reduction in *mec-7* transcript levels we used a translational reporter P*mec-7*::MEC-7::GFP using both cDNA and genomic constructs, along with a touch receptor neuron-specific reporter P*mec-4*::*mCherry* (Fig 6F). We observed diminished absolute (MEC-7::

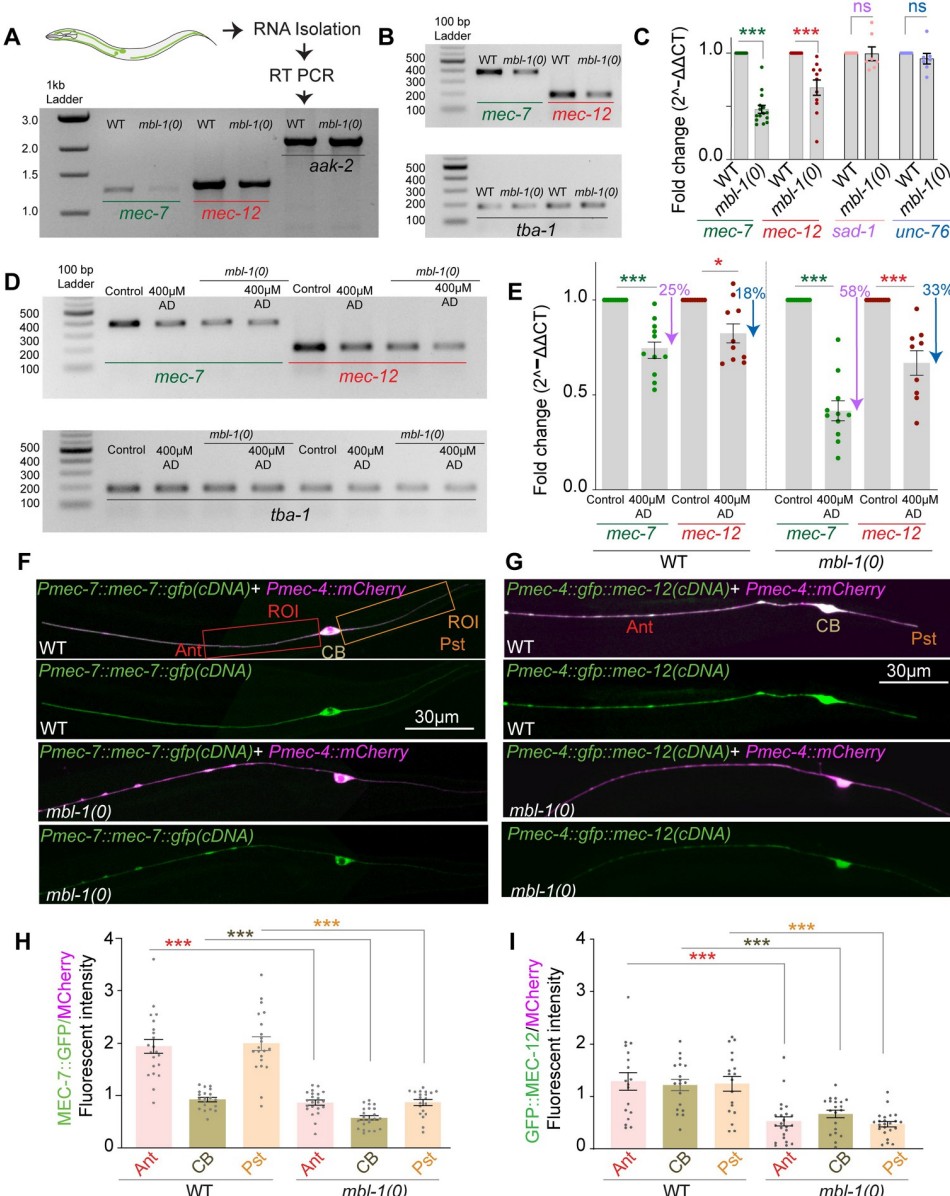

**Fig 6. MBL-1 regulates the levels of *mec-7* and *mec-12* tubulin transcripts.** (A) Illustration depicting the method for reverse transcription (RT)-PCR and representative agarose gel image showing transcript length of *mec-7*, *mec-12*, and *aak-2* (control) in the wild-type and *mbl-1(0)* backgrounds. (B) Representative agarose gel image from the sample of quantitative real-time (qRT) PCR, showing a reduction in the amount of *mec-7* and *mec-12* transcripts in *mbl-1* mutants. *tba-1* has been used as an internal control. (C) Quantification of qRT-PCR of the transcripts of *mec-7*, *mec-12*, *sad-1*, and *unc-76* in the wild-type and *mbl-1(0)* backgrounds. For C, Independent replicates (N) = 10–11 and the number of reactions (n) = 11–15. (D-E) Representative agarose gel (D) and quantification of transcript (E) *mec-7* and *mec-12* in the 400 μM Actinomycin D-treated worms in the wild-type and *mbl-1(0)* backgrounds. The purple and blue arrows depict the percentage drop observed in *mec-7* and *mec-12* transcript levels, respectively, in WT and *mbl-1(0)* on ActD treatment. For E, Independent replicates (N) = 9–10 and the number of reactions (n) = 9–14. (F-G) Representative confocal images of the worms expressing either *Pmec-7::mec-7::gfp (shrEx473)* translation reporter (F) or *Pmec-4::gfp::mec-12 (shrEx492)* translation reporter (G) in wild-type and *mbl-1(0)* backgrounds. The diffusible control reporter *Pmec-4::mCherry (tbIs222)* was used in both (F) and (G). (H) Quantification of ratio (MEC-7::GFP/ MCherry) of fluorescent intensity from 50 μm regions of interest (ROI) in the anterior (Ant) and posterior (Pst) neurites of PLM and the PLM cell body (CB). (I) shows the quantification of the ratio (GFP::MEC-12/ MCherry) of average fluorescent intensity for *mec-12* translation reporter done in a similar manner described for (H) above. For H & I, Independent replicates (N) = 3–4 and the number of neurons (n) = 18–25. For C, E, and H-I, *P < 0.05; ***P < 0.001; ANOVA with Tukey's multiple comparison test. Error bars represent SEM. ns, not significant.

GFP) and normalized (MEC-7::GFP/MCherry) intensity in the anterior and posterior PLM neurites and cell body of *mbl-1* mutants, indicating decreased protein levels of MEC-7 (Fig 6F and 6H), (S6A, S6B, S6F and S6I Fig). A similar trend was observed when both MEC-7::GFP and MCherry were expressed from a single construct P*mec-7*::MEC-7::GFP::*SL2*::MCherry (S6G, S6H and S6J Fig). This suggests that the visible drop in MEC-7::GFP intensity is not due to reduced transcription under a specific promoter. The level of a translational reporter for *mec-12*, GFP::MEC-12, was also significantly reduced in the *mbl-1* mutants (Fig 6G and 6I, S6C and S6D Fig). Based on these results, we concluded that MBL-1 controls the levels of *mec-7* and *mec-12* mRNAs in PLM neurons, potentially by regulating their transcript stability.

Given the observed reduction in tubulin levels in the *mbl-1* mutant and the earlier findings of increased microtubule instability, we hypothesized that microtubule dynamics might also be compromised in *mec-7* and *mec-12* mutants. Supporting this hypothesis, imaging of the EBP-2::GFP reporter in *mec-7* and *mec-12* mutants, revealed a significant increase in the number of EBP-2 tracks in the respective kymographs (S7B and S7D Fig). Additionally, microtubule polarity in the anterior neurite became mixed in these mutants (S7B and S7F Fig), similar to the observations in the *mbl-1* mutants (Fig 3B–3F). These findings further strengthen the notion that tubulin levels are compromised in the *mbl-1* mutant animals leading to disruptions in the microtubule cytoskeleton.

## MBL-1 interacts with *mec-7*, *mec-12* and *sad-1* mRNAs

To determine the interaction between MBL-1 and *mec-7/mec-12* mRNAs, we conducted Ribonucleoprotein Immuno-Precipitation (RIP) using transgenic worms expressing MBL-1::GFP::FLAG under its native promoter. An anti-Flag antibody was used for immuno-precipitation (Fig 7A, left panel). As a control, we used animals expressing KLP-7::GFP::FLAG under its native promoter [70]. The immuno-precipitated sample showed the presence of MBL-1::GFP::FLAG (Fig 7B, red arrow, left panel). We observed the enrichment of *mec-7*, *mec-12*, and *sad-1* transcripts in the immuno-precipitated (IP) sample compared to the control sample (Fig 7C and 7D). However, we did not find any enrichment of *unc-76* mRNA upon immunoprecipitation of MBL-1(Fig 7C and 7D), indicating the specificity of interaction between MBL-1 and *mec-7*, *mec-12*, and *sad-1* transcripts. To further support the association of MBL-1 protein with *mec-7*, *mec-12*, and *sad-1* mRNAs in mechanosensory neurons, we immuno-precipitated GFP-tagged MBL-1 from *mbl-1(0)* animals expressing MBL-1::GFP (C isoform) specifically in the mechanosensory neurons under the P*mec-4* promoter (P*mec4*::MBL-1::GFP) (Fig 7A and 7B, panels on right, for both A and B). P*mec4*::GFP was used as a control. We observed an enrichment of *mec-7* and *sad-1* mRNA, but not *mec-12*, in the immunoprecipitated MBL-1 (C isoform) complex compared to the control (Fig 7D and 7E). This might account for the relatively poor binding of *mec-12* transcripts to the c-isoform of MBL-1.

## Expression of tubulin gene *mec-7/ mec-12* and *sad-1* rescues the 'short neurite' and 'ectopic synapse' phenotype respectively in PLM neurons

In this study, we observed that the *mbl-1* mutant exhibited a short anterior neurite phenotype and reduced amount of tubulin transcripts (*mec-7* and *mec-12*) in mechanosensory neurons. Based on these findings, we hypothesized that increasing *tubulin* transcript in touch neurons in the *mbl-1* mutants could suppress its short neurite phenotype. Indeed, when *mec-7* cDNA was expressed under its native promoter in the *mbl-1* mutant, the proportion of PLM neurons with a short anterior neurite decreased from 80% to 20% (Fig 8A and 8B). Consistently, overexpression of *mec-7* in *mbl-1(0)* mutants using a genomic *mec-7* construct also significantly suppressed this phenotype (Fig 8B). Additionally, overexpression *of mec-7* in wild-type

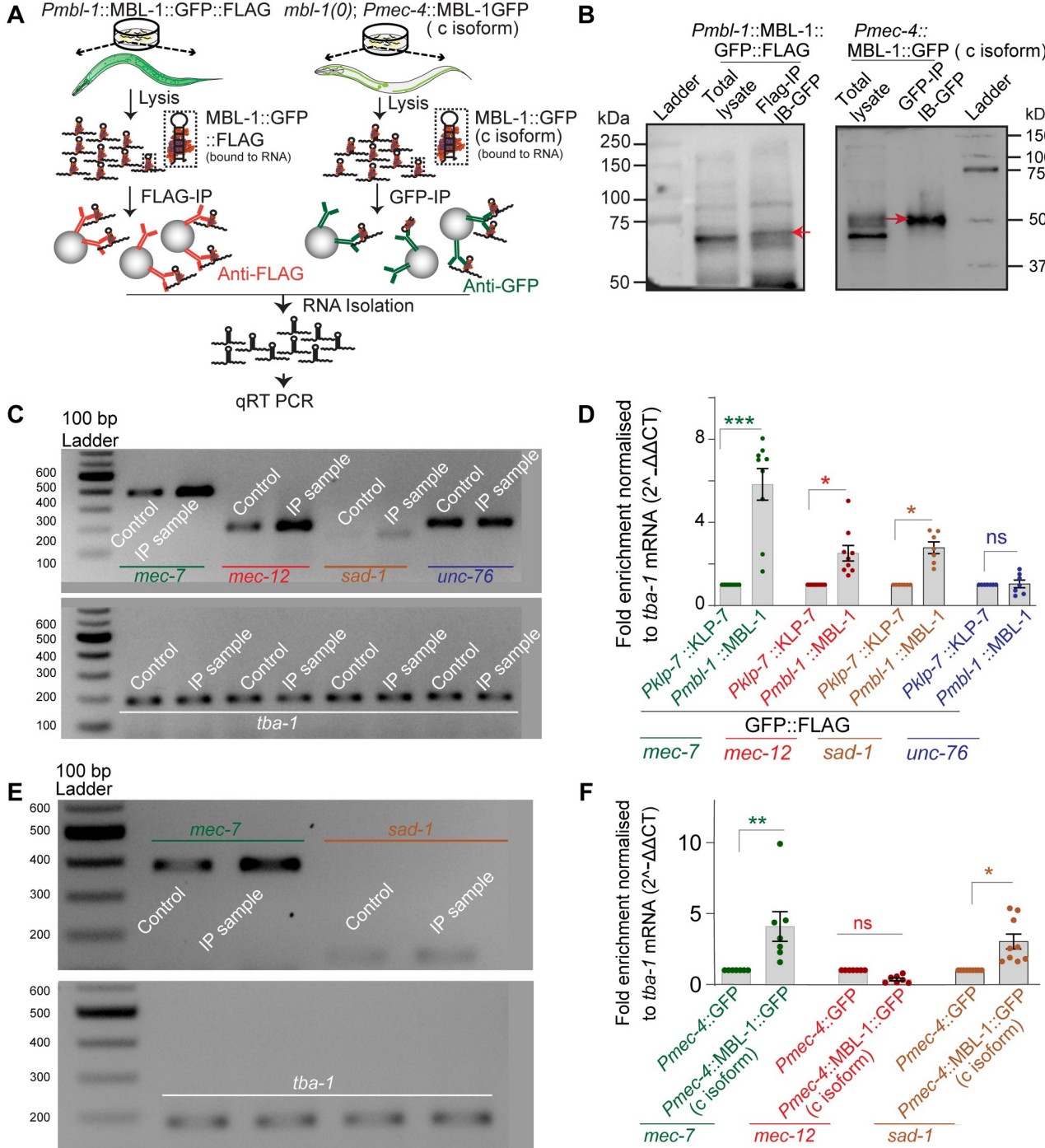

**Fig 7. MBL-1 interacts with *mec-7*, *mec-12* and *sad-1* mRNAs.** (A) An illustration of the Ribonucleoprotein Immuno-Precipitation (RIP-Chip) method and quantitative real-time (qRT) PCR from *Pmbl-1*::MBL-1::GFP::FLAG and *mbl-1(0); Pmec-4*::MBL-1::GFP(C-isoform) backgrounds. (B) Representative western blot picture showing the enrichment of MBL-1::GFP::FLAG and MBL-1::GFP (C isoform) (marked in red arrowhead) in the IP sample as compared to the control sample. (C-D) Representative agarose gel picture (C) and quantification of the transcript (D) of *mec-7*, *mec-12*, *sad-1*, and *unc-76* from the control sample (*Pklp-7*::KLP-7::GFP::FLAG) and experimental sample (*Pmbl-1*::MBL-1GFP::FLAG). *tba-1* was used as an internal control. Independent replicates (N) = 7–9 and the number of reaction (n) = 7–9. (E) Showing the representative agarose gel picture of the transcript of *mec-7* and *sad-1* from qRT-PCR of the control *mbl-1(0); Pmec-4*::GFP (*shrEx481*) and IP sample *mbl-1(0); Pmec-4*::MBL-1::GFP (C-isoform) (*shrEx75*). (F) Bar graph showing the quantification of the transcript of *mec-7*, *mec-12*, and *sad-1* from control (*mbl-1(0); Pmec-4*::GFP *(shrEx481))* and IP sample *(mbl-1(0); Pmec-4*::MBL-1::GFP (C-isoform) (*shrEx75*).

animals resulted in overgrowth of the anterior neurite of PLM (Fig 8A and 8C). Similarly, overexpression of *mec-12* also rescued the short neurite phenotype observed in the *mbl-1* mutant animals (Fig 8A–8C). These findings suggest that tubulin levels are limiting for neurite growth in *mbl-1* mutants. Furthermore, overexpression of tubulins in wild-type worms led to neurite overgrowth, indicating that tubulin levels are sufficient for promoting neurite growth.

We further investigated whether overexpression of tubulins could rescue the gentle touch response defect observed in the *mbl-1* mutant animals. Interestingly, we found that overexpression of both *mec-7* and *mec-12* rescued the anterior touch response in *mbl-1(0)* animals (Fig 8D). However, the posterior touch response was significantly rescued in *mbl-1(0)* mutants only by *mec-7* overexpression (Fig 8E). Based on these observations, we concluded that MBL-1 regulates both neurite growth and the gentle touch response behavior through touch neuron-specific tubulins (*mec-7* and *mec-12*).

In contrast, we did not observe any detectable change in the stability or amount of *sad-1* transcript in *mbl-1(0)* mutants. However, we did observe an enrichment of the *sad-1* transcript in the immunoprecipitated MBL-1 complex. A previous study also indicated that MBL-1 prevents the exclusion of the 15th exon of *sad-1* transcript in ALM neurons, resulting in the expression of only the exon-included isoform [40]. Thus, we speculated that the ectopic synapse defect in *mbl-1(0)* mutants could be attributed to the absence of exon-included isoform of *sad-1* (Fig 8F). Remarkably, this defect was significantly rescued by expressing the exon-included isoform of *sad-1* in the *mbl-1* mutants using a touch neuron-specific promoter (Fig 8G–8F). However, transgenic expression of the genomic *sad-1*, in the *mbl-1* mutants failed to rescue the ectopic synapse defect (Fig 8G). These findings provide further evidence that MBL-1 may be involved in the isoform-specific regulation of *sad-1* transcript, as previously observed [40].

## Discussion

In this study, we uncovered a cell-autonomous role of the RNA-binding protein MBL-1 in neurite outgrowth and synapse formation in touch neurons (Fig 8H). Our findings demonstrated that microtubule stability in PLM touch neurons is compromised in *mbl-1* mutants due to reduced levels of MEC-7 (β-tubulin) and MEC-12 (α-tubulin). Further investigation revealed that MBL-1 directly binds to the *mec-7* and *mec-12* transcripts regulating their stability (Fig 8H). Additionally, *mbl-1* plays a crucial role in the proper positioning of synapses in PLM neurons by regulating *sad-1* (Fig 8H).

### MBL-1 regulates axon growth and synapse formation in neurons

RNA binding proteins (RBP) play important roles in various developmental stages of neurons, including neurogenesis, migration, axon guidance, synapse formation and axon and dendrite outgrowth [71,72]. However, the specific functions of Muscleblind-1 in nervous system development have been less explored. It is known to regulate alternative splicing, localization, stability, and processing of mRNAs [21–23]. In our study, we revealed that the *C. elegans* homolog of MBNL-1, MBL-1, is involved in regulating the neurite growth of PLM neurons and we also observed defects in neurite growth in the BDU interneuron. Previous studies have shown that MBL-1 controls axon guidance by regulating alternative splicing of *Dscam-2* in *Drosophila*, in a cell-autonomous manner [37]. Our data align with the role of the cytoplasmic form of MBNL-1 in promoting neurite extension in primary cultures of mouse hippocampal neurons [36]. Additionally, a previous study in *C. elegans* demonstrated that MBL-1 regulates synapse formation in DA9 motor neurons [39], and we similarly observed that MBL-1 is involved in synapse formation and positioning in PLM neurons. Moreover, we detected significant gaps in

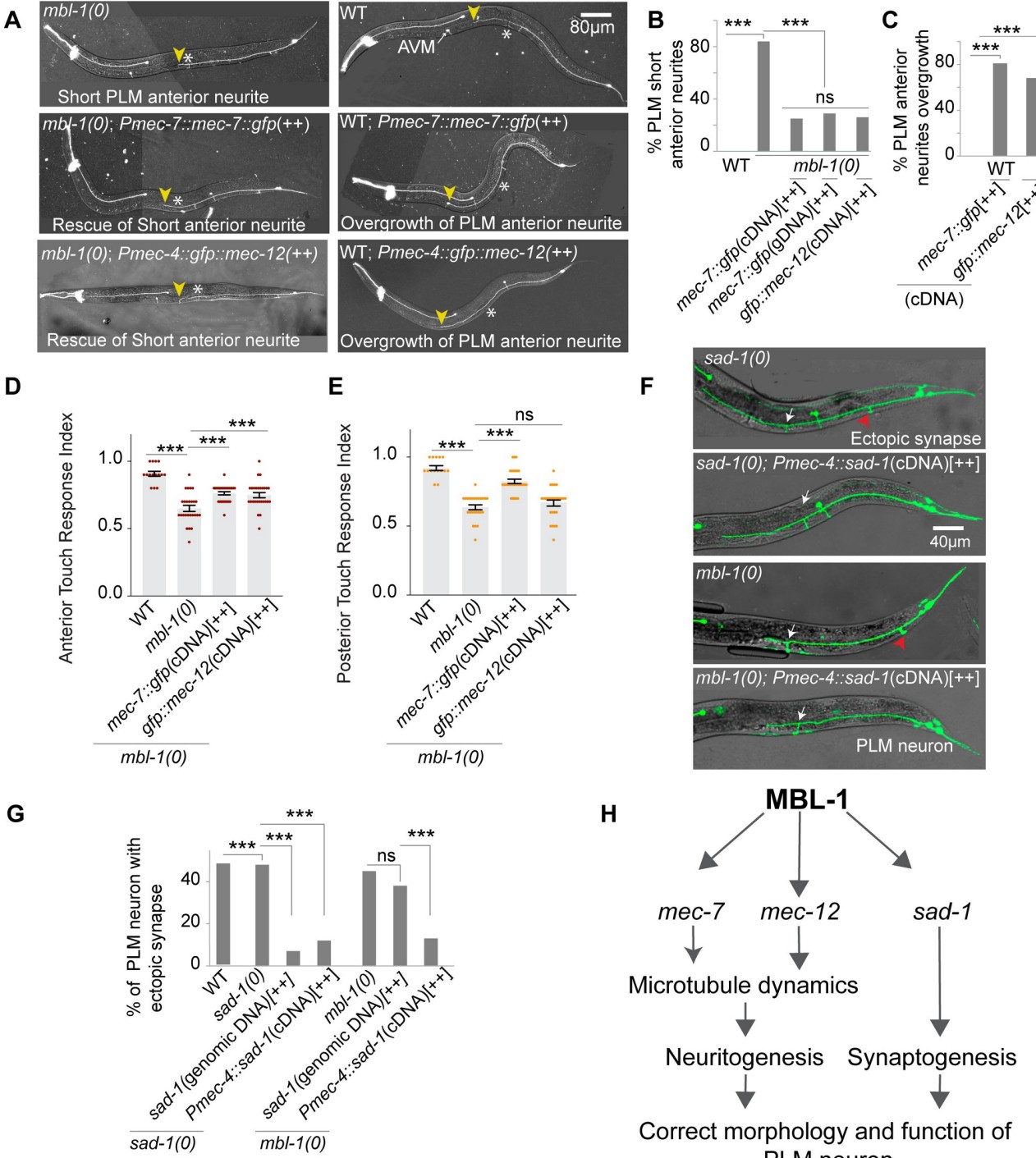

**Fig 8. Overexpression of either *mec-7* or *mec-12* and *sad-1* in the *mbl-1(0)* background rescues short neurite and ectopic synapse phenotype, respectively.** (A) Representative confocal images of PLM/ALM neurons in *mbl-1(0)*, *mbl-1(0); Pmec-7::mec-7::gfp (shrEx475)*[++], *mbl-1(0); Pmec-4:: gfp::mec-12 (shrEx488)*[++], wild-type (WT), WT; *Pmec-7::mec-7::gfp(shrEx475)*[++] *and* WT; *Pmec-4::gfp::mec-12(shrEx489)*[++] background. The yellow arrowhead shows the end of the anterior PLM neurite in *mbl-1(0)*, and the vulval position is marked with an asterisk (*). (B) Histogram is showing quantification of short PLM anterior neurite in different genetic backgrounds. (C) Histogram is showing the overgrowth of PLM anterior neurite in the *Pmec-7::mec-7::gfp* cDNA *(shrEx475)* and *Pmec-4::gfp::mec-12* cDNA *(shrEx489)* in the wild-type background. For B-C, Independent replicates (N) = 3–5 and the number of neurons (n) = 100–145. (D and E) The histogram shows the anterior (D) and posterior (E) gentle touch response index of the worm in the wild-type, *mbl-1(0)*, *mbl-1(0); Pmec-7::mec-7::gfp (shrEx475)*[++], *mbl-1(0); Pmec-4::gfp::mec-12 (shrEx488)*[++] backgrounds. Independent replicates (N) = 3–5 and the number of worms (n) = 25–30. (F) Representative confocal images showing ectopic synapse and

rescue of the ectopic synapse. (G) The bar graph shows the quantification of ectopic synapse in *sad-1(0)*, *sad-1(0); genomic sad-1(shrEx477)* [++], *sad-1 (0); Pmec-4::sad-1(shrEx478)*[++], *mbl-1(0)*, *mbl-1(0); genomic sad-1(shrEx479)*[++] and *mbl-1(0); Pmec-4::sad-1 (shrEx480)*[++] backgrounds. Independent replicates (N) = 3–5 and the number of neurons (n) = 100–200. (H) Illustration showing the model for the regulation of *mec-7*, *mec-12*, and *sad-1* mRNAs by MBL-1 RNA binding protein. For B-C, and G, ***P<0.001, Fisher's exact test. For D-E, ***P < 0.001; ANOVA with Tukey's multiple comparison test. Error bars represent SEM. ns, not significant.

the dorsal cord region of the *mbl-1* mutants using a GABAergic motor neuron reporter, which may indicate improper neurite extension or loss of neuromuscular synapses. In *C. elegans* there are four other, known RNA binding proteins, namely MEC-8/RBMPS, MSI-1/MSI-2, UNC-75/CELF5, and EXC-7/ELAVL4, which control the splicing, stability, and localization of transcripts in neurons [73]. However, loss-of-function mutants of these genes did not exhibit morphological defects in mechanosensory neurons, as observed in the *mbl-1* mutants. Notably, MEC-8 specifically regulates the alternative splicing of *mec-2* in mechanosensory neurons and controls gentle touch sensation [74,75], but we did not observe any morphological defects in PLM neurons upon loss of *mec-8*. It is worth mentioning that patients with Duchenne muscular dystrophy (DMD) exhibit symptoms of progressive muscle degeneration as well as learning disabilities, impaired cognitive function, and memory impairment [76,77]. Similarly, DM1 patients experience an age-related decline in frontotemporal functions, including memory impairment [76,78]. Our results indicate a touch neuron-related behavioral deficit in the *mbl-1* mutant, which is consistent with these observations.

## MBL-1 regulates neuronal microtubule cytoskeleton for controlling axon growth

We initially identified the *mbl-1* mutation as a suppressor of the *klp-7* mutant, which causes ectopic neurite outgrowth due to microtubule hyperstabilization. Our suppressor screen was specifically designed to isolate factors involved in microtubule stabilization. Through this screen, we discovered that the *mbl-1* mutation suppresses the abnormal neurite outgrowth phenotype caused by the *klp-7* mutation, indicating that MBL-1 plays a role in regulating microtubule stability. Although MBNL is primarily known as an RNA-binding protein, its involvement in microtubule cytoskeleton regulation and RNA transport has been emerging [41,79]. A thorough investigation of neuronal microtubule dynamics in the *mbl-1* mutants revealed an increase in microtubule instability as well as compromised microtubule orientation in the anterior neurite of the PLM neurons. A similar observation has been reported in the mutants of *mec-17*, an α-tubulin acetyltransferase, that regulates the microtubule stability and axonal integrity of PLM neurons [80]. The increased microtubule instability correlates with the mixed polarity of microtubules, which is also observed in the posterior neurite of the PLM neurons [7]. Similarly, in vertebrate dendrites, microtubules are relatively unstable compared to the axon and display random orientation [3,10].

In the *mbl-1* mutant, along with the strong effect on microtubule stability, we observed a severe reduction in the anterograde transport of the synaptic protein RAB-3. These findings are consistent with previous studies demonstrating that MBL-1/2 knockout in mice cortical neurons leads to reduced dendritic complexity and alterations in postsynaptic densities [81]. The observed changes in axonal and dendritic morphology have been linked to cytoskeletal machinery [82,83]. Through *in-silico* analysis, immunoprecipitation, and RT-PCR, we demonstrated that MBL-1 directly binds to the transcripts of both α (*mec-12*) and β-tubulins (*mec-7*), regulating their transcript levels. However, we did not detect a specific association between *mec-12* and the c-isoform of MBL-1 in mechanosensory neurons, which could be attributed to the weak interaction of the MBL-1 c-isoform with *mec-12* mRNA. *In-silico* analysis revealed

five different consensus sequences recognized by MBNL-1/MBL-1, each with different binding probabilities for the recognized transcripts, including *sad-1*, *mec-7*, and *mec-12* (S4B Fig). Mutations in *mec-7* and *mec-12* lead to defects in microtubule dynamics and orientation like those observed in the *mbl-1* mutants. Notably, overexpressing tubulins (*mec-12* and *mec-7*) in the *mbl-1* mutants rescued both the neurite-growth defect and touch response defect. These findings are consistent with previous reports describing the roles of touch neuron-specific tubulins (*mec-7/12*) in microtubule regulation and neurite outgrowth [47–49]. Our data suggest that MBL-1 is involved in regulating the stability of α (*mec-12)* and β (*mec-7)* tubulin transcripts for axon growth in PLM neurons. However, the mechanisms underlying the regulation of *mec-7* or *mec-12* transcript stability by MBL-1 remain unclear. MBL-1 is known to act as an adaptor for mRNA localization [23,84]. The stability of tubulin transcripts may be achieved through such mechanisms.

Previous studies have shown that MBL-1 regulates the splicing of *sad-1* in ALM and BDU neurons of *C. elegans* [40]. The function of SAD-1 in synapse formation and stabilization is well established [6,85]. We observed ectopic synapse formation in both *mbl-1(0)* and *sad-1(0)* mutants indicating that these two genes act in the same pathway to ensure proper synapse formation. The *sad-1* has two isoforms, each with a cell-specific expression pattern [40]. When we over-expressed the exon-fifteenth included isoform specifically in touch neurons in the *mbl-1(0)* background, it suppressed the ectopic synapse phenotype. However, over-expression of the genomic *sad-1* failed to suppress the ectopic synapse phenotype. These results support the hypothesis that MBL-1 regulates the splicing of *sad-1* to control synapse formation. In summary, this study provides mechanistic insights into how an RNA-binding protein, MBL-1, regulates the structure and function of neurons through the cytoskeletal machinery (Fig 8H).

## Materials and methods

### *C. elegans* genetics

*C. elegans* strains were cultured on standard Nematode Growth Medium (NGM) plates seeded with OP50 *Escherichia coli* bacteria at 20˚C [86]. All the loss-of-function mutant alleles were denoted as "*0*". For instance, the *mbl-1(tm1563)* mutant was represented as *mbl-1(0)*. The N2 Bristol wild-type strain was used to eliminate background mutations, while CB4856 Hawaiian isolates were employed for restriction fragment length polymorphism (RFLP) mapping. The list of mutant strains and transgenic reporter strains used in this study can be found in S5 and S6 Tables, respectively. Additionally, the list of newly generated transgenes, carrying extrachromosomal arrays utilized in this study is included in S6 Table.

Transgenes were introduced into the various mutant backgrounds through crossing. PCR or sequencing methods were employed to confirm the homozygosity of all mutants. The following published transgenes were utilized in this study, *Pmec-7*::GFP (*muIs32*), *Pmec-4*::EBP-2::GFP (*juIs338)* [87], *Pmec-4*::MCherry (*tbIs222)* [88], *Pmec-7*::GFP::RAB3 (*jsIs821)* [53], and *Pmec-7*::TagRFP::ELKS-1 (*jsIs1075*) [54].

### EMS mutagenesis and forward genetic screen strategy

We conducted an EMS mutagenesis screen [86] to isolate suppressor mutants, which exhibited suppression of the ectopic neurite growth phenotype in the ALM neurons of *klp-7(0)* animals. Age-synchronized L4 stage *klp-7(0)* animals were pooled and subjected to three washes with 1XM9 solution. Subsequently, the animals were incubated in 1XM9 media containing 47mM EMS at 20˚C for 4 hours with continuous mixing. After the incubation period, we pelleted the animals and discarded the supernatant, followed by three additional washes with 1XM9 solution.

Next, we transferred approximately 6–7 healthy late L4 stage animals to fresh NGM plates seeded with OP50 bacteria. From these parent plates, around 700 F1 progeny animals were single-selfed at the L4 stage onto fresh seeded NGM plates. We then screened the resulting F2 progeny to identify mutants exhibiting the suppression of ALM ectopic extension phenotype. We isolated three suppressor mutants, one of which, *ju1128*, was mapped to the *mbl-1* gene, as discussed in the paper.

## Mapping of *ju1128* mutation by restriction fragment length polymorphism (RFLP)

To map the *ju1128* mutation, we crossed the *klp-7* suppressor strain carrying the *ju1128* mutation (*klp-7(0); ju1128*) with the polymorphic *C. elegans* Hawaiian strain (CB4856). The individual worms from the resulting F2 progeny were single-selfed on NGM plates and their progeny was allowed to grow. The progeny from each of these plates was then genotyped to identify the presence of *klp-7(tm2143)* deletion mutation. The F2 plates with the confirmed *klp-7(tm2143)* genotype were subsequently phenotyped to assess the suppression of ALM ectopic extension phenotype, and therefore identify homozygotes for suppressor mutation.

We selected 30–50 F2 plates that were homozygous for both *klp-7(tm2143)* and suppressor. To determine the chromosomal location, we pooled DNA from these 30–50 unique F2 recombinants and performed SNP mapping. This involved using three primers per chromosome for ChrI-ChrV and two primers for the X-chromosome, following previously established protocols [89]. The results of the SNP mapping are presented in S1A Fig. Since the EMS suppressor screening was conducted in the N2 (Bristol) strain, we utilized N2 SNPs to establish the linkage.

In the SNP mapping gel image, we observed linkage between N2 SNPs and two chromosomes: the 3rd chromosome and the X chromosome (marked with a red arrowhead in S1A Fig). As the *klp-7* gene is known to reside on the 3rd chromosome, we inferred that the linkage observed on this chromosome was associated with the *klp-7* mutation. Additionally, the linkage observed on the X chromosome was likely due to the *ju1128* mutation (marked with a red arrowhead in S1A Fig). This inference was consistent with the results of whole-genome sequencing data analysis.

## Whole-genome sequencing analysis for *ju1128* mapping

The same F2 recombinants used for SNP mapping, as discussed in the previous section, were also utilized for whole genome sequencing analysis. We extracted genomic DNA from these recombinants using the phenol-chloroform extraction and ethanol precipitation method. The DNA samples were then sent for sequencing on an Illumina HiSeq4000 platform, generating 50 bp paired-end reads.

After filtering out low-quality reads, a total of 300 million high-quality reads were retained, resulting in an average coverage of 18X for the *C. elegans* genome from this dataset. These reads were aligned to the *C. elegans* reference genome version WS220, and subsequent analysis was performed using the CloudMap pipeline [51]. By analyzing the aligned reads, a single file containing all the variations was obtained using genome-wide variant call statistics.

To identify candidate mutations specific to *ju1128* allele, background variations from the parental strain *klp-7(0)* and other sister mutants isolated in the same screen (such as *ju1130*) were subtracted from the list of total variants. This process yielded a filtered list of candidate mutations. The resulting list was further annotated using the available reference annotation file of *C. elegans*.

From this analysis, eight candidate genes, including the *mbl*-1 gene, were short-listed for the *ju1128* mutation. To validate the potential involvement of these candidate genes, we performed rescue experiments by injecting a fosmid carrying the wild-type copy of each gene into *klp-7(0); ju1128* background. We observed rescue of ectopic neurite suppression phenotype by the fosmid carrying the *mbl-1* gene in *klp-7(0); ju1128* background (Fig 1A).

## Widefield fluorescence imaging of mechanosensory neurons for quantifying developmental defects

The touch receptor neurons (TRNs) were phenotyped on a Leica DM5000B microscope under a 40X objective (NA 0.75). For all imaging purposes, animals at L4 stage were immobilized and mounted on 5% agarose beds using 10 mM levamisole (Sigma-Aldrich; L3008) in 1XM9 buffer. The morphological defects in ALM neurons resulting from loss of *klp-7* (Fig 1A and 1C) or in PLM neurons resulting from the loss of *mbl-1* (Fig 2A and 2B) were assessed qualitatively based on their microscopic appearance. The posterior neurite of ALM neurons was considered as an ectopic extension if it extended to or beyond the vulva in animals at the L4 stage (Fig 1A and 1C). Similarly, the anterior neurite of PLM was considered a 'short neurite' if it did not extend beyond the vulva. This method was also used to determine the percentage of worms with ectopic synapses or more than one synapse phenotype in the *mbl-1(0)* background (Fig 2C–2E).

## Image acquisition and analysis of neurite length of mechanosensory neurons using a point-scanning confocal microscope

The ALM/PLM neurons expressing a fluorescent reporter under a tissue-specific promoter, *Pmec-7*::GFP (*muIs32*), were imaged in L4 stage animals using a Zeiss Axio Observer LSM 510 Meta confocal microscope. The imaging was performed at 66% of a 488 nm laser under a 40X oil objective (NA 1.3). Simultaneously, the differential interference contrast images of these animals were captured.

The absolute length of the anterior and posterior neurites of PLM, as well as the anterior neurite of ALM, was calculated using either Zeiss LSM Image Browser software or ImageJ. These software tools were used to draw segmented traces over the neurite in a fluorescent channel image and measure the length value of the neurite.

The length value of the anterior neurite of PLM was normalized to the distance between PLM cell body and vulva of the corresponding animal, measured from the differential interference contrast images (Fig 1E). The length of the posterior neurite of PLM was normalized to the distance between PLM cell body and tip of the tail (Fig 1D). In the same manner, the length of the anterior neurite of ALM was normalized to the distance between vulva and tip of the head (Fig 5E), as previously described [49].

## Image acquisition for GFP::RAB-3, *Punc-86*::GFP, *Punc-25*::GFP, ELKS-1:: TagRFP, and UNC-9::GFP using a point-scanning confocal microscope

The transgenic animals expressing GFP::RAB-3, *Punc-86*::GFP, *Punc-25*::GFP, ELKS-1:: TagRFP, and UNC-9::GFP, were imaged using a Nikon A1HD25 confocal microscope under 60X oil objective (NA 1.4). The GFP reporter was imaged using 7% of the 488 nm laser. The TagRFP and MCherry reporters were imaged using 0.3% of the 561 nm laser.

## Molecular cloning and generation of new transgene

In order to express *mbl-1* under pan-neuronal, touch neuron-specific, and muscle-specific promoters, we first made an expression gateway entry clone for *mbl-1* pCR8::*mbl-1*(cDNA)/ pNBRGWY29. The *mbl-1* cDNA (c-isoform) was obtained from Yuji Kohara collection, in Japan [90]. The pNBRGWY29/*pCR8*::*mbl*-1(cDNA)) was LR recombined with pCZGY66 (*Prgf-1* destination vector), pCZGY553 (*Pmec-4* destination vector), and pCZGY61 (*Pmyo-3* destination vector) respectively, using LR clonase enzyme (Invitrogen;11791–020). To make *Pmec-4*::*sad-1* (pNBRGWY164), the entry clone pNBR58 corresponding to *sad-1a* cDNA was recombined with pCZGY553 (*Pmec-4* destination vector). *sad-1* was amplified using the following primers 5'TCCGAATTCGCCCTTCGTCAATCGGGCAAAGTC 3'and 3'GTCGAATTCGCCCTTGATGATAGATTAGACTTTATCAGCC 5'and was cloned into pCR8 vector using an infusion reaction (Takara, 638947).

The *mec-7* constructs *Pmec-7*::*mec-7*::*gfp* (cDNA) (pNBR165) and *Pmec-7*::*mec-7*::*gfp* (genomic) (pNBR166), were constructed by an LR reaction between the destination vector *Pmec-7*:: GWY::GFP (pNBR61) and *mec-7* genomic and cDNA entry clones. The gateway construct was the result of another infusion reaction between *mec-7* promoter, linearized using 5'CCAT- GATTACGCCAATGGCGCGCCAAATGTAAACC 3'and 3'TGGCCAATCCCGGGGC- GAATCGATAGGATCCACGATCTCG 5'primers, and GWY::GFP vector backbone, linearized using, 5'CCCCGGGATTGGCCAAAG 3'and 3'TTGGCGTAATCATGGTCA- TAGCTG 5'primer pair. The *mec-7* cDNA and genomic DNA fragments were amplified using the following primers: 5'TCCGAATTCGCCCTTATGCGCGAGATCGTTCATATTC 3'and 3'GTCGAATTCGCCCTTCTCTCCGTCGAACGCTTC 5'. Subsequently, each was cloned into the pCR8 vector backbone using infusion reactions to make entry clones *mec-7* cDNA (pNBR59) and *mec-7* genomic (pNBR60).

*Pmec-7*::*mec-7*::*gfp*::*SL2*::*mCherry* (genomic) (pNBR66) was constructed by linearizing the vector backbone *Pmec-7*::*mec-7*::*gfp* (genomic) (pNBRGWY166) using following primers 5' GATATCTGAGCTCCGCATCG 3' and 3' GCATGGACGAGCTGTACAAGTAA 5' and amplifying the insert *SL2-mCherry* from *Pmec-4*::*Chrimson*::*SL2*::*mCherry*::*unc-54 (*Addgene Plasmid #107745) using the following primers 5' GAGCTGTACAAGTAAGCTGTCT- CATCCTACTTTCACC 3' and 3' GGTGGCATGGATGAATTGTATAAGATATCT- GAGCTCCG 5'. The two linearized fragments were then recombined by an infusion reaction (Takara, 638947).

To make *Pmec-4*::*gfp*::*mec-12* (cDNA) (pNBRGWY170) construct, pCR8::*mec-12* (cDNA) (made by Topo cloning, (Thermo Fisher Scientific; K2500-20)) was single-site LR recombined with pCZGY1867 (*Pmec-4*::*gfp* destination vector) (Takara, 638947). The following primers were used for amplifying *mec-12* cDNA: 5'CTCTTTTGCAAAATGAGAGAAGTAATTTCG 3'and 3'GAGAAGAAGGAGATGAGTATTAGGCG 5'.

These plasmids were injected at different concentrations as described in S6 Table. *Pttx-3*:: RFP was used as a coinjection marker at 50 ng/ul concentration to generate transgenic strains. The total DNA concentration of the injection mixture was kept at around 110–120 ng/μl by adding pBluescript (pBSK) plasmid to the injection mixture.

## Live imaging of EBP-2::GFP, and GFP::RAB-3 using spinning disk confocal microscopy

For imaging EBP-2::GFP and GFP::RAB-3 we used a Zeiss Observer Z1 microscope equipped with a Yokogawa CSU-XA1 spinning disk confocal head and a Photometric Evolve electron-multiplying charge-coupled device camera for fast time-lapse image acquisition.

EBP-2::GFP imaging was conducted for a total duration of 2 minutes at a rate of 2.64 frames per second. For the GFP::RAB-3 imaging, images were acquired at a rate of 3.19 frames per second for a total duration of 3 minutes. To achieve the best signal-to-noise ratio, EBP-2::GFP was imaged with an excitation laser of 8.75mW at 488nm. While a power of 10mW was utilized for GFP::RAB-3 imaging.

### Analysis of EBP-2::GFP and GFP::RAB-3 dynamics

The kymographs of EBP-2::GFP (Fig 3B) and GFP::RAB-3 (Fig 4B) were generated using the Analyze/ MultiKymograph tool in ImageJ software (https://imagej.nih.gov/ij/). To create these kymographs 30 μm regions of interest (ROIs) were placed on both the anterior and posterior neurite of PLM (Figs 3A and 4A). In both types of kymographs, the horizontal axis represents the neurite length in micro-meters, and the vertical axis represents the duration of time in seconds.

The microtubule dynamics was quantified from EBP-2::GFP movies by generating kymographs from the 30 μm ROIs. The ROIs were drawn on the anterior neurite of PLM in a distal-to-proximal direction (towards the cell body) and on the posterior neurite in a proximal-to-distal direction (away from the cell body). The diagonal tracks which were moving away from the cell body were annotated "plus-end-out" microtubules (P; green traces in Fig 3B), while diagonal tracks moving towards the cell body were denoted as "minus-end-out" microtubules (M; purple traces in Fig 3B). The fraction polarity of microtubules was calculated by determining the number of plus-end out tracks or minus-end out tracks out of the total number of tracks in each EBP-2::GFP kymograph. The growth length and the growth duration of EBP-2:: GFP tracks was calculated as a net pixel shift in the X and Y axes, respectively (Fig 3E and 3F).

The GFP::RAB-3 movies were analyzed to study axonal transport. Similar to EBP-2::GFP movies, ROIs were placed on the anterior and posterior neurite of PLM to generate kymographs. The diagonal tracks moving away from the cell body were annotated "anterograde" (green traces in Fig 4B), while diagonal tracks moving towards the cell body were denoted as "retrograde" (purple traces in Fig 4B). We calculated anterograde and retrograde particle movement from each kymograph by quantifying the number of tracks in either the anterograde or retrograde direction from the 30 μm ROIs corresponding to the anterior and posterior neurite of PLM near the cell body during the 3 minutes of imaging. We calculated run length by quantifying the net pixel shift in the x-axes in the anterograde and retrograde directions [53].

### Gentle touch assay

The L4 stage hermaphrodite worms were subjected to a gentle touch assay using the tip of an eyelash, following the protocols described previously [88,91]. Each animal was given ten alternative touches at the anterior and posterior ends. A response was considered positive if the touch elicited a reversal behavior. A positive response was denoted as 1 and no response as 0. To quantify the touch response, we calculated the anterior touch response index (ATRI) (Fig 4G) and the posterior touch response index (PTRI) (Fig 4H) as a ratio of the total number of responses to the total number of touches given (which was 10 touches per animal).

### Identification and analysis of MBL-1 targets

To identify MBL-1 targets in the PLM neuron, we first used the CeNGEN database [56] to list all the genes expressed in PLM neurons, considering a threshold value of 2. It enlisted a total of 5,283 genes which are expressed in the PLM neuron (S1 Table) (CeNGEN database).

Next, we employed the oRNAment database (http://rnabiology.ircm.qc.ca/oRNAment) to identify potential genes with MBL-1 binding site (CGCU) out of 5283 genes expressed in PLM neurons. This analysis resulted in the identification of 2000 genes that potentially contain MBL-1 binding sites and exhibit enriched expression in the PLM neurons.

To further narrow down our focus, we conducted gene ontology (GO) analysis and short-listed genes involved in the following processes: (1) Microtubules-based (2) axon development (3) regulation of synapse structure (4) Axodendritic transport. The results of this analysis are presented in Fig 5A and S2 Table.

### Reverse transcription (RT) PCR for checking the splicing of the transcript

To examine splicing defects in the *mbl-1(0)*, we used the reverse transcription method. The total RNA was extracted from wild-type and *mbl-1(tm1563)* age-synchronized worms at the day-one adult stage (A1). To age synchronize gravid adults were allowed to lay eggs for 30 minutes and then the progeny was grown at 20°C till the 1 day old adult stage. The synchronized animals were washed three times with M9 buffer, and the pellet was collected and stored at -80°C for RNA isolation. RNA was isolated from thawed pellet using the Qiagen RNeasy Mini Kit (no. 74104; Qiagen). The extracted RNA was treated with DNase I (Ambion's DNA-free kit AM1906) to remove any genomic DNA contamination. Approximately 3–4 μg of the treated RNA was reverse transcribed into cDNA using Superscript III Reverse Transcriptase (18080093). For splicing analysis, we used 200 ng cDNA from either wild-type or *mbl-1 (tm1563)* background in a 25 μl PCR reaction. Emerald Amp GT PCR (2X master mix, cat-RR310) Taq polymerase was used for the PCR amplification of *mec-7*, *mec-12*, *sad-1*, and *unc-76* transcripts. The sequences of the specific primers used are given in S3 Table, and the primer binding sites are depicted in S5A Fig. For *mec-7* and *mec-12* transcripts, primers were designed to amplify the full length of the transcript. These genes have only one isoform based on Worm-base-WS285 annotation. The amplification was performed for 30 cycles. The 5 μl volumes of PCR products of *mec-7*, *mec-12*, and *aak-2* (used as a control) were separated on a 1% agarose gel (Fig 6A), while *sad-1* and *unc-76* PCR products were separated on a 2% agarose gel (S6B Fig).

### Quantitative real-time (qRT-PCR) for checking the total transcript

We used 1 day old adult animals to isolate the total RNA from wild-type and *mbl-1(0)* mutant backgrounds, as described in the previous section. The extracted RNA was treated with DNase I (Ambion's DNA-free kit AM1906) to get rid of any genomic DNA contamination. Subsequently, approximately 2–3 μg DNase-treated RNA was reverse transcribed into cDNA using Superscript III Reverse Transcriptase (Invitrogen no. 18080093).

For each qRT reaction, 50ng of this cDNA was added to 20 μl of Power SYBR Green PCR Master Mix (Applied Biosystems Life Technologies, no. 3367659). The amplification was performed for 40 cycles.

The primers designed for this experiment were selected to ensure a single Ct peak for each qRT PCR reaction, thereby ensuring the specificity of the amplicon being quantified. Secondly, to prevent amplification of any residual genomic DNA contaminants, some primers were designed with binding sites at the intron-exon boundary. The primer sequences and their relative positioning are given in the supplementary information in S4 Table and S6A Fig, respectively.

Gel electrophoresis of PCR products on a 3% agarose gel, did not show any contaminating bands. The relative mRNA levels of target genes in the *mbl-1(tm1563)* and the wild-type N2 strains were calculated using the standard ΔΔCt method [92]. The calculated values were

normalized to *tba-1* values, which served as a control for endogenous mRNAs [93]. We used the ΔΔCT method for calculating the fold change of transcript relative to respective controls in the wild-type and *mbl-1(tm153)* background [92].

## Transcriptional inhibition experiment

The RNA synthesis was inhibited by feeding animals 400 μM actinomycin-D (Sigma, A9415), dissolved in DMSO, as previously described [94,95]. OP-50 bacteria were cultured in B-broth media overnight at 37°C in a BOD incubator. Actinomycin D solution in DMSO was further diluted in OP-50 B-broth to prepare a working concentration of 400 μM, for control, the same volume of DMSO was dissolved in OP50. These ActD and DMSO dissolved cultures were then used to seed a 60 mm NGM plate. Gravid adult animals were transferred on NGM plates containing ActD or DMSO for 30 minutes to allow for egg laying and age synchronization. One day old adult animals, which were grown on these plates, were then washed thrice with 1XM9, collected, and frozen at -80°C. These pellets were later thawed on ice to isolate RNA for qRT PCR as described above.

## Ribonucleoprotein-immuno precipitation (RIP)

Immunoprecipitation experiments were conducted following previously described protocols [96]. Approximately 50–100 gravid adult worms expressing either *Pmbl-1*-MBL-1::GFP::FLAG [55,97] or P*mec-4-mbl-1*::*GFP* (c-isoform) were transferred to thirty 60 mm NGM plates and allowed to lay eggs for half an hour at 20°C. The resulting progeny was then grown until the one day old adult stage (A1) at 20°C. These synchronized animals were then pooled, washed three times with 1XM9, and centrifuged at 1500 rpm for 2 minutes. The collected pellet, which was more than 300 μl in volume, was then stored at -80°C until further use. For the RIP experiment, the pellet was thawed on ice for homogenization and all subsequent procedures were performed at 4°C. The animal pellets were homogenized in 400–500 μl of ice-cold 2X lysis solution [which consisted of Buffer A (20 mM Tris (pH 8.0), 150 mM NaCl, 10 mM EDTA) + 1.5 mM DTT, 0.2% NP-40, 0.5 mg/ml Heparin Sulphate, 1X EDTA complete Protease inhibitor (Roche -11836153001, 1 tablet for 5ml), RNase inhibitor (Invitrogen AM2696, 50 U/ml), Phosphatase inhibitor (100 U/ml) and RNase out (Invitrogen 1643272, 100 U/ml)].

The homogenized sample was passed successively through 19 mm, 22 mm, 26 mm, and an Insulin syringe to make a smooth homogenate, which was then centrifuged, at 19,000Xg for 20 minutes, to obtain a clear supernatant. A portion (5–10%) of the total lysate was set aside as the total input for RNA estimation and western blotting. The total protein concentration in each sample was determined using the Bradford assay. Equal amounts of protein were used across all conditions for the RIP experiments.

Agarose beads (Roche, 11719416001) were equilibrated in 1X lysis buffer. For pre-clearing, each supernatant was incubated with 20 μl of equilibrated agarose beads for one hour, followed by centrifugation at 8,000Xg for 10 minutes to collect the beads. The supernatant obtained after this step was the pre-cleared lysate, which was further processed.

To immunoprecipitate Flag-tagged GFP from the strains, anti-Flag M2 agarose beads (Sigma, A2220-1ML) were used. For each sample 60 μl of 50% slurry (equivalent to 30 μl of packed beads) of anti-Flag M2 agarose beads was taken in a 1.5 ml tube and washed with 1X lysis buffer to equilibrate. The pre-cleared lysates from the previous step were added to the equilibrated beads and incubated at 4°C overnight with continuous mixing. In the case of immunoprecipitation of *Pmec-4*::MBL-1::GFP from wild-type and *mbl-1* samples expressing this transgene, 3 μg of anti-GFP antibody (MBL LifeSciences M048-3, raised in mouse) was added to the precleared lysates and incubated for 8–10 hours with continuous mixing. The

following day, 30 μl of equilibrated Protein G–agarose beads (Sigma 15920010) were added to the lysates and incubated further for 3 hours.

Both samples (anti-Flag M2 and anti-GFP) were then centrifuged at 10,000 rpm for 15 minutes at 4˚C. The resulting pellets were collected and washed with lysis buffer containing 150 mM NaCl, followed by centrifugation at 10,000Xg for 15 minutes at 4˚C. The agarose beads were collected, with 20% of the bead's volume reserved for western Blot analysis and the remaining 80% was used for RNA isolation and qRT PCR as described earlier.

## Western blot

Samples were boiled in Laemmli buffer and resolved on a 12% SDS-PAGE. After the proteins were transferred to nitrocellulose membranes, the membranes were blocked with 5% BSA for one hour and then probed with the anti-GFP antibody (Abcam ab290) overnight. The following day, the membranes were washed in 1X Tris Buffer Saline containing 0.1% Tween 20 (TBST) and then incubated with a secondary antibody (anti-Rb) for 3 hours. Finally, the protein bands were detected using a standard ECL chemiluminescence detection kit (Millipore, WBKLS0500).

## Image Acquisition and Quantification for *Pmec-7*::MEC-7::GFP *and Pmec-4*::GFP::MEC-12

For image acquisition and quantification of fluorescent intensities a Nikon confocal microscope (A1HD25) with a 60X oil objective (NA 1.4) was used. The worms were mounted on 5% agarose beds and immobilized using 10 mM levamisole in M9 buffer. The transgenic worms co-expressing *Pmec-4*::*mCherry* (*tbIs222*) along with either *Pmec-7*::*mec-7*::*gfp* (cDNA, *shrEx473)* or *Pmec-7*::*mec-7*::*gfp* (genomic DNA, *shrEx474)*were imaged using 10% of 488 nm laser for GFP tagged cDNA, 1.5% of 488nm laser for GFP tagged genomic DNA, and 0.3% of 561 nm laser for mCherry (constitutive reporter).

To analyze the fluorescent intensity, average fluorescent intensities of *mec-7*::*gfp*, or *mec-12*::*gfp*, and *mCherry* reporters were measured from the PLM cell body and the 50 μm regions of both the anterior and posterior neurite of PLM neuron, using ImageJ. We measured average intensities from ROIs placed on the background, outside the PLM neuron for background correction. The ratio of average intensities of GFP and MCherry (Fig 6F) as well as the absolute intensities of GFP and MCherry (S6A and S6B Fig) were plotted based on these measurements.

Similarly, worms expressing *Pmec-7-mec-7* (genomic DNA)::*gfp*::*SL2*::*mCherry*, which co-expresses the *mec-7*::*gfp* (genomic DNA) and *mCherry* under the same promoter,) were imaged using 8% of 488 nm laser for GFP and 1% of 561 nm laser for mCherry from the same ROIs as discussed above.

For quantifying the ratio of the average intensities of *gfp*::*mec-12* and *mCherry* (Fig 6F) and their absolute fluorescent intensities (S7C and S7D Fig) worms coexpressing *Pmec-4*::*gfp*::*mec-12* (cDNA, *shrEx492)* and *Pmec-4*::*mCherry* (*tbIs222*) were imaged using 5% of 488 nm laser for GFP::*mec-12* and 1% of 561 nm laser for mCherry.

## Statistical analysis

The data analysis was performed using GraphPad Prism software version 9.0.2. The mean value and the standard error of the mean (SEM) were represented by the bar in the plots. The $X^2$ test (Fisher's exact test) was used for comparing proportions, while ANOVA with a post hoc Tukey's multiple comparisons test was used for comparing more than two groups. Bartlett's test was conducted to test the homogeneity of variances before proceeding with ANOVA.

The P-value, indicating the level of significance, was presented in each panel of the Figure to compare the respective groups. The sample number (n) for each experiment was mentioned in the respective Figure legend, along with the total number of biological replicates (N).

## Supporting information

**S1 Fig. Mapping of the *ju1128* mutation, related to Fig 1.** (A) The Gel picture showing the result of Restriction Fragment Length Polymorphism (RFLP) mapping of *ju1128* mutation. The 100-base pair ladder was used as a marker. In the gel picture, mutation showed linkage on two chromosomes, one on the third chromosome, which is because of *klp-7(tm2143)* (marked with red arrowhead) and another on the X chromosome (marked with red arrowhead). (B) Frequency plot of X-chromosome for mapping of the *ju1128* mutation from the whole genome sequencing data using the method described by Minevich et al. (2012), with chromosome position (in megabases/Mb) plotted against the Normalized frequency of pure parental alleles. It peaks at the genomic position linked to the *ju1128* mutation.
(TIF)

**S2 Fig. *mbl-1* mutants display a defect in the BDU neuron and GAP junction, related to Fig 2.** (A) Representative confocal images and (B) illustration of touch neurons (ALM and PLM) and BDU neurons in both wild-type and *mbl-1(0)* at the L4 stage. BDU neurons were visualized using *Punc-86*::GFP (*kyIs262*) and for visualization of touch neurons, *Pmec-4*::mCherry (*tbIs222*) and *Punc-86*::GFP (*kyIs262*) transgenes were used. The presence of physical contact between the PLM anterior neurite and BDU neuron is shown by the white arrowhead in the wild-type background which is lost in *mbl-1(0)* shown by the red arrowhead. (C) Quantification of the defect as shown in the image (A). N = 3 independent replicates, n (number of worms) = 25–32. (D-E) Representative confocal images (D) and schematics (E) of the gap junction synapse labeled with UNC-9::GFP*(shrEx434)* in the touch neurons, in the wild-type and *mbl-1(0)* backgrounds. The green arrows show the localization of UNC-9::GFP in the wild-type background, whereas in the *mbl-1(0)* the red arrow is pointed at the tip of PLM anterior neurite missing UNC-9::GFP localization. (F) Quantification of the percentage of defect shown in the image in panel D. N = 3–4 independent replicates, n (number of worms) = 30–35. (G-I) Representative confocal images (G) and illustration (H) of PLM neuron in the wild-type (*muIs32)* and *mbl-1(tm1563); muIs32* backgrounds at A7 (seven-day adult) stage. White arrows are showing the visible gaps in the anterior neurite indicating degeneration. (I) Histogram showing the quantification of degeneration of PLM neurons at A3 (three-day adult stage), A5 (five-day adult stage), and A7 in wild-type and *mbl-1(tm1563)* background. N = 3 independent replicates, n (number of worms) = 16–26. For C, and F***P <0.001; Fisher's exact test. ns, not significant.
(TIF)

**S3 Fig. *mbl-1* mutants display a defect in D-type GABAergic motor neurons, related to Fig 2.** (A-C). Confocal images of wild type (A) and *mbl-1(0)* (B) worms expressing a presynaptic reporter *Pmec-7*-ELKS-1::TagRFP (*jsIs1075*), shown in magenta color. The neuron is also labeled with diffusible reporter *Pmec-7*-GFP (*muIs32*), shown in green. The ectopic synapse in the PLM anterior process in the *mbl-1(0)* background is marked by a red arrowhead, whereas the original synapse is marked by a white arrowhead. (C) The histogram shows the percentage of PLM neurons with ectopic synapses in the *mbl-1(0)* background. N = 3 independent replicates, n (number of worms) = 25–30. (D-F) Representative images (D) and schematic (E) of D-type motor neurons labeled with P*unc-25*::GFP (*juIs76*) in the wild-type and *mbl-1(0)* at the L4 stage. Red arrow showing a defect in neurite growth in *mbl-1(0)*

background. (F) The histogram shows the percentage of worms showing neurite defects in the *mbl-1(0)*. N = 3–4 independent replicates, n (number of worms) = 30–50. (G) Representative confocal images of touch neurons in the mutants of different RNA binding proteins. The names of the mutants are mentioned next to the respective image panels. (H) Quantification of any defects in PLM morphology seen in the mutants mentioned in (G) N = 3 independent replicates, n (number of worms) = 80–130. For C, F, and H ***$P < 0.001$; Fisher's exact test. ns, not significant.
(TIF)

**S4 Fig. Putative MBL-1/MBNL-1 binding sites in different transcripts, related to Fig 5.** (A-D) Pictures depicting MBL-1/MBNL-1 preferential binding sequence and binding positions in the transcript of *mec-7* (A), *mec-12* (B), *unc-76* (C), and *sad-1* (D). (E and F) The histograms show the anterior (E) and posterior (F) gentle touch response index of the worm in the wild-type, *mbl-1(0)*, *mec-7(0)*, *mec-12(0)*, *mbl-1(0) mec-7(0)* and *mbl-1(0); mec-12(0)* backgrounds. N = 3 independent replicates, n (number of worms) = 16–50. For E-F, **$P < 0.01$; ***$P < 0.001$. Error bars represent SEM. Statistical comparisons were done using ANOVA with Tukey's multiple comparison test. ns, not significant.
(TIF)

**S5 Fig. Relative quantification of different transcripts in *mbl-1(0)* background using quantitative RT-PCR, related to Fig 6.** (A) Illustration showing the positions of different primers, used for checking the transcript length or doing qRT-PCR, on *tba-1*, *mec-7*, *mec-12*, *sad-1*, and *unc-76* genes. The sequence of these primers is given in the supplementary file S4 Table. (B) Representative agarose gel image showing *sad-1* and *unc-76* transcript in the wild-type and the *mbl-1(0)* background. (C) The histogram is showing the relative fold change of the transcript of *mec-7*, *mec-12*, *sad-1*, and *unc-76* in the wild type and the *mbl-1(0)* backgrounds. These data were obtained from quantitative real-time PCR (qRT-PCR). Independent replicates (N) = 6 and the number of reaction (n) = 6–8. For C, ***$P < 0.001$; ANOVA with Tukey's multiple comparison test. Error bars represent SEM. ns, not significant.
(TIF)

**S6 Fig. Quantification of fluorescent intensity of MEC-7::GFP and GFP::MEC-12 reporters in wild-type and *mbl-1(0)* background, related to Fig 6.** (A-D) Histograms showing the absolute fluorescence intensities of *Pmec-7::mec-7::gfp* (cDNA) *(shrEx473)* (A-B), *Pmec-4::gfp::mec-12* (cDNA) *(shrEx492)* (C-D), and *Pmec-4::mCherry (tbIs222)* (A-D) in the wild-type and *mbl-1(0)* backgrounds. The fluorescence intensity is quantified in an arbitrary unit from the anterior (Ant) and posterior (Pst) neurites of PLM, from 50 μm regions of interest (ROI) as shown in the Fig 6F, and the PLM cell body in the wild-type and *mbl-1(0)* backgrounds. For A-D, independent replicates (N) = 3–4 and the number of neurons (n) = 20–25. (E and G) Representative confocal images of the worms expressing *Pmec-7::mec-7::gfp* (genomic DNA) *(shrEx474)* and *Pmec-4::mCherry (tbIs222)* (E), and worms expressing *Pmec-7::mec-7::gfp::SL2::mCherry (shrEx486)* (G) in wild-type and *mbl-1(0)* background. (F and H) The histogram is showing quantification of the ratio (MEC-7::GFP/MCherry) of average fluorescent intensity from 50 μm regions of interest (ROI) as shown in Fig 6F in the anterior (Ant) and posterior (Pst) neurites and cell body (CB) of PLM neurons. (I-J) Histograms showing the fluorescence intensities of *Pmec-7::mec-7::gfp* (genomic DNA) *(shrEx474)* and *Pmec-4::mCherry (tbIs222)* in arbitrary units (I) and histogram showing fluorescence intensities of *Pmec-7::mec-7::gfp::SL2::mCherry (shrEx486)* (J) in the wild-type and *mbl-1(0)* backgrounds. For F, H, I, and J, independent replicates (N) = 3–4 and the number of neurons (n) = 20–25. For A-D, F, H, and I-J, *$P < 0.05$; **$P < 0.01$; ***$P < 0.001$; ANOVA with Tukey's multiple comparison test. Error

bars represent SEM. ns, not significant.
(TIF)

**S7 Fig. *mec-7(0)* and *mec-12(0)* mutants display a defect in the microtubule dynamics, related to Fig 6.** (A) Schematic of the PLM neuron showing the 30 μm Regions of interest (ROIs), marked in red and orange for anterior and posterior neurites, respectively. These ROIs were used for analyzing the time-lapse movies of *Pmec-4*::EBP-2::GFP (*juIs338*) in wild-type, *mec-7(0)*, and *mec-12(0)* backgrounds. (B) Representative kymographs of EBP-2:: GFP obtained in the wild-type, *mec-7(0)*, and *mec-12(0)* backgrounds obtained from the above-mentioned ROIs. The green and magenta traces on kymographs represent the Plus-end-out (microtubule growth events away from the cell body) and Minus-end-out (towards the cell body) tracks, respectively. (C) The bar graph is showing the fraction of microtubules with plus-end-out' (P) or 'minus-end-out' (M) polarity in wild-type, *mec-7(0), and mec-12 (0)* backgrounds in the PLM anterior (Ant) and posterior (Pst) neurites. N = 3–5 independent replicates, n (number of worms) = 20–50. (D) The bar graph represents the number of EBP-2::GFP tracks (number of growing microtubules) in PLM anterior (Ant) and posterior (Pst) neurites in wild-type, *mec-7(0), and mec-12(0)*. N = 3–5 independent replicates, n (number of worms) = 20–50. (E and F) Growth length (E) and growth duration (F) of the tracks, measured from net pixel shift in the X and Y axis, respectively, from kymographs shown in B. N = 3–5 independent replicates, n (number of tracks) = 1986–6064. For C-F, ***P < 0.001; ANOVA with Tukey's multiple comparison test. Error bars represent SEM. ns, not significant.
(TIF)

**S1 Table. Transcripts with MBL-1 binding site expressed in PLM neuron.**
(XLSX)

**S2 Table. GO analysis of MBL-1 targets involved in different biological processes.**
(XLSX)

**S3 Table. MBL-1 targets checked for a phenotype in PLM neuron.**
(XLSX)

**S4 Table. Primers used for doing qRT-PCR.**
(XLSX)

**S5 Table. Strains used in this study.**
(XLSX)

**S6 Table. Transgenes Generated.**
(XLSX)

## Acknowledgments

We would like to thank the National Bio-Resource Project, Japan, and the Caenorhabditis Genetics Centre for strains. We thank Andrew Chisholm and Yishi Jin for their support and guidance at the initial stage of this project. The *ju1128* mutant was isolated in their labs. We thank Neeraj Singh and Pankajam Thyagarajan for their assistance in WGS data analysis. We thank Sibaram Behera for helping with touch assay experiments and for the help in Bioinformatics analysis. We thank Arnab Mukhopadhyay, Sandhya Koushika, Michael Nonet, and Cori Bargmann for their help with strains. We are also grateful to Yuji Kohara for sharing *mbl-1* cDNA.

## Author Contributions

**Conceptualization:** Dharmendra Puri, Sunanda Sharma, Sourav Banerjee, Anindya Ghosh-Roy.

**Formal analysis:** Dharmendra Puri, Sunanda Sharma, Sarbani Samaddar, Sruthy Ravivarma.

**Funding acquisition:** Sourav Banerjee, Anindya Ghosh-Roy.

**Investigation:** Dharmendra Puri, Sunanda Sharma, Sarbani Samaddar, Sruthy Ravivarma, Anindya Ghosh-Roy.

**Methodology:** Dharmendra Puri, Sunanda Sharma, Anindya Ghosh-Roy.

**Project administration:** Anindya Ghosh-Roy.

**Resources:** Anindya Ghosh-Roy.

**Supervision:** Sourav Banerjee, Anindya Ghosh-Roy.

**Validation:** Dharmendra Puri, Sunanda Sharma, Sarbani Samaddar.

**Visualization:** Dharmendra Puri, Sunanda Sharma, Sarbani Samaddar.

**Writing – original draft:** Dharmendra Puri, Sarbani Samaddar, Anindya Ghosh-Roy.

**Writing – review & editing:** Dharmendra Puri, Sunanda Sharma, Sarbani Samaddar, Sourav Banerjee, Anindya Ghosh-Roy.

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
