## [Decision Letter · Decision Letter 0]

31 Oct 2022

Dear Dr. Ghosh-Roy,

Thank you very much for submitting your Research Article entitled 'Musleblind-1 regulates microtubule cytoskeleton in *C. elegans* mechanosensory neuron through tubulin mRNAs' to PLOS Genetics.

The manuscript was fully evaluated at the editorial level and by three independent peer reviewers. The reviewers appreciated the attention to an important problem, but raised some substantial concerns about the current manuscript that need to be addressed to provide mechanistic insights, expand the data analysis, and strengthen the conclusions.  These are especially related, but not limited, to the insufficient data provided to support the conclusion that MBL-1 stabilizes the *mec-7* and *mec-12* transcripts, how *mbl-1* suppresses *kpl-7*, and what link exists between the regulation of *mec-7/12* by *mbl-1* and the microtubule polarity and dynamics.  Moreover, the manuscript will need to be structured within the context of previous published work, some of which has not been cited.  This includes the early work on neuronal development and polarization, as indicated by Reviewer 1, as well as the work on *mbl-1* in synaptic formation(Shen lab) that should be presented as the entry point for this study of *mbl-1* on synapses.  Finally, the manuscript will need to be carefully checked for grammatical errors, typos, and colloquial language, which populate the current version.  Based on these reviews, we will not be able to accept this version of the manuscript, but we would be willing to review a much-revised version. We cannot, of course, promise publication at that time.

If you decide to revise the manuscript for further consideration at PLOS Genetics, please aim to resubmit within the next 60 days, unless it will take extra time to address the concerns of the reviewers, in which case we would appreciate an expected resubmission date by email to plosgenetics@plos.org.

We are sorry that we cannot be more positive about your manuscript at this stage. Please do not hesitate to contact us if you have any concerns or questions.

Yours sincerely,

Massimo A. Hilliard

Guest Editor

PLOS Genetics

Gregory P. Copenhaver

Editor-in-Chief

PLOS Genetics

Reviewer's Responses to Questions

**Comments to the Authors:**

Reviewer #1: The article by Puri et al. investigates the role of MBL-1 in neuronal structure and synapse localization in the mechanosensory neurons of C. elegans. It provides new insights into the importance of this molecule for correct neuronal structure. It also offers mechanistic insights by demonstrating interactions with mec-7, mec-12, and sad-1 mRNAS. Thus, overall it provides novel data on how an RNA binding protein interacts with cytoskeletal elements to mediate correct neuronal structure. As listed below, I have numerous suggestions for the authors to address before the manuscript can be considered for publication.

Overall, the writing of the manuscript needs significant improvement. The grammar and sentence structure need to be improved to help with the readability of the manuscript, as there are many instances in which the meaning of the authors is unclear.

It would be important to cite some of the early work on neuronal development and polarization in C. elegans in the introduction. For example, Adler et al., 2006 Nat Neuro; Chang et al., 2006 Curr Biol; Hilliard and Bargmann, 2006 Dev Cell; Prasad and Clark, 2006 Development; Quinn et al., 2008 Curr Biol.

All gene names should be italicized.

Figure 1.

Panel A. The authors may wish to label the PVM neurons in their images to make them more understandable for those who are not familiar with the mec neurons of C. elegans.

Panel C. Was the ALM defect analysed with a yes or no qualification, ie it either had a posterior neurite or it didn’t? This is not clear in the manuscript.

Panel D and E. The stats are not clear in these two graphs – is the fosmid rescue line significantly different to the controls (ie WT and mbl-1(ju1128)?

Is the fosmid able to rescue the tm1563 allele?

Figure 2.

Why did the authors choose to focus on the tm1563 allele for this figure (for images and rescue) when they primarily used the ju1128 allele in Figure 1?

Panel A. In the image of the mbl-1 mutant, the ALM and PLM axons appear much fainter than the WT and possibly fragmented. Is there degeneration occurring in these neurons? Thus, is the short phenotype developmental or is it degenerative?

Panel B. It’s interesting that the TRN-specific rescue is not as strong as the genomic or pan-neuronal rescues. Why do the authors think this is so? Is this difference significantly different?

Panel F. The text states that “MBL-1::GFP was 238 highly enriched in many neurons including ALM and PLM touch neurons”. However, from the images provided it is impossible to see the PLM neurons. It appears that the authors have performed the required co-localization experiments using a mCherry marker, but this is not visible in these images. Please provide clearer and/or enlarged images and/or quantification to demonstrate that MBL-1 is expressed in the ALM and PLM neurons. Furthermore, do you find enrichment of MBL-1 in any areas of these neurons?

Figure 3.

The author may wish to consider separating this figure into two to make it easier to follow.

Panel M/N. The functional decline in the posterior response is quite subtle. It would be good to show that this is rescued with the introduction of the rescue transgene to provide further strength the findings. Expanding subsequent findings, especially with mec-12, mec-7 and sad-7, to assess their effects on the mbl-1-mediated reductions in touch response would provide further strength to the proposed interactions and into their mechanisms of action.

Figure 4.

Both mec-7 and mec-12 mutants have previously been shown to present a high penetrance of axonal degeneration in the mec neurons. However, here the authors again refer to the phenotype as short neurites. The authors need to include additional experiments that clarify their phenotype as either developmental or degenerative.

Figure 5.

Panel D-E. I’m not sure that I follow the authors assumption that “This reduction observed in the mbl-1(0) mutants as compared to the wild type is interesting, as it illustrates the fate of the pre-existing transcripts in the absence of MBL-1 and in the absence of any new transcription. As transcription was blocked, the observed additional decrease could be attributed to the increased instability of the pre-existing transcripts in the absence of MBL-1.” It seems that the more plausible explanation would be that this experiment is showing the normal turn-over of the transcripts, as for both WT and mutant you show an ~40% decrease after treatment. Thus, the mutant is appearing to behave the same as the WT after treatment. As such, I am not convinced by the conclusion that “MBL-1 is regulating the stability of mec-7 and mec-12 mRNA in PLM neurons.”

Panel F/G. Did the authors find similar results with mec-12?

Figure 6.

Panel B/C. These panels show interaction with mec-7, mec-12 and sad-1, but then mec-12 appears to be forgotten about for the next experiments in this figure and in the text, which only refers to mec-7 and sad-1. Why is this? Did the authors find that mec-12 did not interact with MBL-1 in the mec neurons in subsequent experiments? The experiments conducted on mec-7 and sad-1 in panels D-E should also be included for mec-12.

Figure 7.

Again, why wasn’t mec-12 included in these analyses? It is included in the model in panel F. Although MEC-12 and MEC-7 are most commonly known as the major subunits of microtubules in the mec neurons, there are several examples in the literature demonstrating that they can function quite differently to one another and influence specific phenotype in different ways or to different degrees. Thus, it cannot be assumed that they will behave the same if the experiments have not been performed.

Panel F. The authors may wish to expand their model to make it more informative and reflective of their proposed mechanisms of action.

Figure S2. The images are far too small to allow the reader to see the required detail. Yes, the schematics are of great assistance, but the original images need to be able to be seen.

The text states “In the mbl-1(0) mutant, we noticed that on the dorsal side, there are often

gaps (red arrowhead, Figure S2J-L) indicating a synaptic defect”. However, the authors have only used a diffuse GFP marker to visualise these neurons. Further analysis using a synaptic marker in these neurons is needed to conclude that these gaps indicate a synaptic defect rather than shorter neurites (similar to the phenotype in ALM/PLM).

Reviewer #2: Review_PGENETICS-D-22-01039

Puri, Ghosh-Roy et al. described how C. elegans MBL-1/Muscleblind, an RNA-binding protein, promotes axon growth and synapse development by regulating microtubule polarity through controlling the level of tubulin mRNAs. They first identified mbl-1 as a suppressor for touch neuron defects in the klp-7 Kinesin mutant, followed by providing evidence that microtubule dynamics and levels of touch neuron-specific tubulins (mec-7, mec-12) are altered in the mbl-1 mutant. RNA-IP experiments showed that MBL-1 binds mec-7 RNA, suggesting that MBL-1 sustains MEC-7 levels through a posttranscriptional mechanism. Consistent with this notion, overexpression of mec-7 rescued the neurite defects of the touch neurons in the mbl-1 mutant. Most of the experiments in this study were reasonable, although some clearly need further refinement in terms of data analysis. However, it remains unexplored how regulation of tubulins by MBL-1 is linked to microtubule polarity and dynamics in the touch neurons. The interaction between mbl-1 and sad-1 is not explored at a deeper level, making one wonder whether the molecular mechanisms by which mbl-1 regulates tubulins and sad-1 are similar or distinct. Such unaddressed issues compromise the mechanistic depth of the paper.

Major points:

1. The paper needs a thorough editing check to correct the numerous grammatical errors and typos in writing. The authors should be more careful in checking their submitted manuscript. In addition, the authors have a tendency of using terms that are either too casual, or seem alienated to the audience without explanation. For example, in line 311 and line 314, “mode value” and “fraction polarity value” were used to describe the polarity defects of the mbl-1 mutant. I suggest the authors revise all terms like these to make their paper better received by the audience.

2. The suppression of klp-7 mutant phenotypes by mbl-1 remains unexplained. The authors need to test the following: (1) Whether microtubule dynamics in the klp-7 are different from the WT and the mbl-1 mutant, and whether defects of microtubule dynamics in the klp-7 mutant are suppressed by the mbl-1 mutation; (2) Whether mec-7 and mec-12 levels are altered by the klp-7 mutation, and whether these defects are again corrected by adding the mbl-1 mutation.

3. What is the link between mec-7/mec-12 regulation by mbl-1 and intact microtubule polarity and dynamics in the touch neurons? As had been stated above, this is the most important mechanistic issue for the current study that needs to be rigorously explored by experiments.

4. What is the basis of axon defects in the D-type motor neurons of the mbl-1 mutant? Obviously, this is unlikely to be the consequence of altered mec-7 and mec-12 levels, as these two tubulin genes are not expressed in DD or VD neurons. Can the authors speculate from the candidate MBL-1 bound RNAs (Figure 4A) and elaborate on this issue (by discussion or experiments)?

5. Figure 5D, 5E: The interpretation is flawed that instability of mec-7 and mec-12 transcripts is increased in the mbl-1 mutant. Since the expression of these two genes is already low in the mbl-1 mutant, the authors need to normalize the levels of mec-7 and mec-12 transcripts after Actinomycin D to those at baseline, and compare this value to that of the wild type animals. Without such quantification, the current conclusion that mbl-1 promotes the stability of mec-7 and mec-12 transcripts is premature and unsupported.

6. Figure 5F, 5G, Line 467-472: I could not find statements of using Actinomycin D in this experiment, so I assumed that it was conducted without Actinomycin D. If so, then the data presented here cannot be used to support the role of MBL-1 in maintaining MEC-7 stability. Moreover, the cytosolic mCherry protein was expressed from a different transgene using a different promoter (Pmec-4) from the mec-7-expressing transgene driven from the mec-7 promoter. As such, comparing the GFP to mCherry ratio using these two transgenes is not a valid way of normalizing MEC-7::GFP quantity. The authors need to create a Pmec-7::MEC-7::GFP::SL2::mCherry transgene for this purpose. That said, these experiments only explore the role of MBL-1 in controlling MEC-7 level without specifically examining its function in promoting MEC-7 stability, which clearly requires the use of Actinomycin D. These experiments need rigorous revision.

7. Figure 6: A negative control for the RNA-IP is necessary to show that MBL-1 does not bind most RNAs in a non-specific fashion. Using the KLP-7::GFP as a control is fine, but the authors need to show that not all RNAs are enriched in the RIP samples of MBL-1::GFP worms.

Minor points:

1. Line 47 and 107: please explain sad-1 (such as “sad-1/SAD BRSK kinase”)

2. Figure 1B: It is better to just show the gene structure of mbl-1 and what the ju1128 and tm1563 alleles are, rather than showing the fosmid contents.

3. The authors are suggested to provide high power images for the neurite and synaptic branch defects of the mbl-1 mutants as well as MBL-1::GFP in Figure 2. It is hard to see the details of the ALM and PLM phenotypes in the current images (Figures 2A, 2C, 2D and 2F).

4. Line 122: The term “multiple axon-like phenotype” (of ALM) is a bit confusing. Although there is still debate whether this is a bipolar ALM phenotype or a misguidance defect of the ALM process, the term that the authors used does not help clarify the issue. I suggest that they just say “ectopic ALM process” (to be exact, the long process of ALM is not a pure axon – it displays features of dendrites).

5. Line 182: Please correct this sentence: “making a ventral synaptic branch the PLM axon makes a synapse…”

6. Line 192: “Pmec-4” should be italicized

7. Line 242: Gaps in the dorsal axons of D-type motor neurons should be interpreted as defects in axon extension similar to the short ALM and PLM neurites, rather than as synaptic defects. The authors should examine the distribution of RAB-3 and ELKS-1 in the D-type motor neurons of the mbl-1 mutant to substantiate their conclusion that mbl-1 is required for synaptic development in this class of neurons.

8. Line 246-247: Please annotate these genes succinctly.

9. Line 252: correct “caused due”

10. Figure 3: The green and purple lines in embedded in Figure 3B are very hard to see. Figure 3J: Some lines are skewed.

11. Line 263: “%” should be “percentage”.

12. Line 268: “smaller” should be “shorter”.

13. It is not clear what microtubule “arrangement” means in the paper. Microtubule “polarity” or “orientation” could be a better term, or the authors need to provide convincing justification for using such a term.

14. Line 382-387: I am confused here as the authors said that they found mec-7 and mec-12 were among genes found to be downregulated by mbl-1 in some published reports but claimed that they could not find genes in this group that were related to GO terms of axon development or synapse structure. Aren’t mec-7 and mec-12 two such genes?

15. Line 420: The expression of “0.4717 +/- 0.03720 fold” is very strange and inappropriate. This could be seen again in Line 456. The authors should carefully revise their use of scientific expression. As has been stated above, this manuscript needs a thorough editing of writing. The senior author should take the responsibility to ensure the quality of writing.

16. Figure 6F, left: The protein band for MBL-1::GFP seems to be absent in the total lysate of the MBL-1::GFP::FLAG animals. The authors need to exlain this. A minor point is that the protein bands in the total lysates of the MBL-1::GFP::FLAG and Pmec-4::MBL-1::GFP look quite different. Why?

17. Figure 7A, the third image from top: That should be the AVM neuron, not the AVA neuron.

Reviewer #3: Puri et al. reveal the role of mbl-1 in neurite and synapse biology through regulation of microtubule cytoskeleton components. In a genetic screen for suppressors of ectopically extended neurites in the ALM neuron in C. elegans, they identify a mutation in mbl-1 that shortens neurites in klp-7 mutants that have enhanced microtubule stability. mbl-1 mutants displayed mixed microtubule polarity and defects in axonal transport of cargo. MBL-1 is an RNA-binding protein, and the authors show evidence that it regulates the mRNAs of the microtubule genes mec-7 and mec-12, as well as sad-1. The authors propose that MBL-1 binds these mRNAs and stabilises them or mediates splicing in the case of sad-1. Overall, the manuscript is technically robust, novel, and interesting, however some of the mechanistic data into MBL-1 function needs improving as does the explanation of results that do not fit the conclusions made. There are some outstanding questions that also remain unaddressed.

Major:

1. The interpretation that MBL-1 stabilises the mRNA molecules of mec-7 and mec-12 is based on insufficient data. While the use of Actinomycin D is good, it may not block transcription fully and the authors need to prove whether this is indeed the case in their animals. Moreover, the translational MEC-7::GFP reporter cannot distinguish between transcriptional or post-transcriptional mec-7 regulation. Therefore, other orthogonal evidence is required to conclude that MBL-1 stabilises these specific transcripts.

2. Some of the data do not fit the conclusions. From the RIP experiments, MBL-1 associates with mec-7 and sad-1 mRNAs in mechanosensory neurons, but not mec-12. In light of this, how do you reconcile its regulation of mec-12 in PLM? Sad-1 mRNA levels or splicing are not altered in mbl-1(0) mutants, yet it associates with MBL-1 protein. If association results in mRNA stability or splicing, why aren’t sad-1 mRNA levels lower or altered in length in animals lacking mbl-1?

3. There are also some unanswered questions. Why do PLM anterior processes develop a mixed microtubule polarity if mec-1 and mec-12 mRNA levels are reduced in mbl-1(0) mutants? Why is the number of growing microtubules increased in mbl-1(0) mutants?

MINOR.

At times, the manuscript is written in a descriptive manner (phenotypic data) and could be improved with more contextualisation and/or insight into the functional consequences of the observed neurite changes. Highlighting the conceptual advances or the point of each result throughout the manuscript may also improve readability in some sections.

Figure 2B: Authors should clarify in the main text and legend that TRN refers to mechanosensory neurons under pmec-4.

Page 8 – line 183 – only photos for one of the mutant alleles is shown in fig. 2A.

Page 8 – line 198 – The UNC-9::GFP reporter appears as puncta throughout the body of the wild-type animal in fig. S2D. Many of these puncta are missing in the mbl-1(0) mutant. Can the authors explain this? In addition, it is also very difficult to understand how the schematics in panel E of this figure were derived from the representative images. For example, there are many UNC-9::GFP puncta along the wild-type PLM axon that are not included in the schematic. What are these? This result needs better clarification.

Page 15 – line 351 – According to Table S3, 22 genes are noted are assessed, rather than 13.

Page 21 –line 469 - missing mcherry transgene nomenclature.

Page 15 -lines 342 – 362 should be condensed. It reads as repetitious.

Figure 4 - Authors should indicate what fluorescent reporter is used for the imaging and analysis. Figure needs images of controls/wild types for reference.

Some grammatical and sentence structure errors. Some parts of the text could be simplified to avoid confusion. The protein of interest is spelt incorrectly in the title.

**Have all data underlying the figures and results presented in the manuscript been provided?**

Reviewer #1: Yes

Reviewer #2: Yes

Reviewer #3: Yes

PLOS authors have the option to publish the peer review history of their article (what does this mean?). If published, this will include your full peer review and any attached files.

Reviewer #1: No

Reviewer #2: No

Reviewer #3: No

---

## [Decision Letter · Decision Letter 1]

12 Jun 2023

Dear Dr Ghosh-Roy,

Thank you very much for submitting your Research Article entitled 'Muscleblind-1 regulates microtubule cytoskeleton in C. elegans mechanosensory neurons through tubulin mRNAs' to PLOS Genetics.

The manuscript was fully evaluated at the editorial level and by independent peer reviewers. The reviewers appreciated the attention to an important topic but identified some concerns that we ask you address in a revised manuscript.

We therefore ask you to modify the manuscript according to the review recommendations. Your revisions should address the specific points made by each reviewer.

Yours sincerely,

Massimo A. Hilliard

Guest Editor

PLOS Genetics

Gregory P. Copenhaver

Editor-in-Chief

PLOS Genetics

Dear Dr. Ghosh Roy,

Thank you for submitting a revised version of your manuscript. The reviewers have examined the paper and they have found it highly improved compared to the original submission. However, they found that there are still some editing elements that needs to be addressed before the paper can proceed further. Importantly, they also highlighted that the manuscript still needs improvement in the writing, grammar, and sentences structure; please ensure that these also are addressed before your final submission.

Unless you believe additional time is needed, I trust a revised version can be submitted within 2 weeks.

I look forward to receive your revised manuscript.

Thank you,

Kind regards,

Massimo.

Reviewer's Responses to Questions

**Comments to the Authors:**

Reviewer #1: The authors have addressed most of my previous concerns and the paper is now much improved from the first submission. However, I still have some minor concerns that should be addressed prior to the manuscript being published.

1. The writing still needs significant improvement for clarity, and to correct grammatical errors and issues with sentence structure.

2. The title should be revised for clarity and to better describe the study. I suggest the following: "Muscleblind-1 interacts with tubulin mRNAs to regulate the microtubule cytoskeleton in the C. elegans mechanosensory neurons"

3. The authors should provide a definition for the sad-1 gene in the abstract. Currently, no indication is given for what this gene is, and therefore there is no capacity for the reader to grasp the importance of the interaction between MBL-1 and sad-1.

4. The TRN phenotypes of the mbl-1 mutants are remarkably similar to those observed in the absence of MEC-17 (neuronal defects, MT defects, synaptic defects, interactions with MEC-12/MEC-7). Yet, there is no mention of MEC-17 in the manuscript. The authors should at least include a mention of the similarities and potential involvement of MEC-17 in their discussion.

Reviewer #2: The authors had done significant amount of work for the revision, and these additional data strengthened the conclusion and clarified most of the issues that I raised during the initial review of the paper. The revised manuscript was much better than the original one. I think the paper can now be considered for publication at PLoS Genetics, after the following minor points are addressed:

1. sad-1 should be explained in the abstract.

2. In Author summary, the last statement "... might help mitigate the diseases such as muscular dystrophy caused due to misregulation of MBNL in humans" is better removed or tuned down. The study was conducted in neuronal cells, which have very different gene expression patterns and cell biology to those of the muscle cells. The current statement is misleading and exaggerating.

3. Line 119: The anterior cells should include AVM? Or the authors should just remove "anterior cells" and "posterior cells" to avoid confusion. As my comments in the initial review, the authors should avoid using jargons and instead stick with the standard terminology of the worm community.

4. I cannot find description of the suppressor screen in the Materials & Methods section. A few descriptions will be necessary, such as the mutagenesis method (EMS?), an estimation of the screen size (number of haploid genomes screened), etc. By contrast, a detailed description of mapping ju1128 in the Results may not be necessary or should be streamlined (line 136-147).

5. Line 175: "mbl" should be italicized.

6. Line 205: It should read "In both mutant alleles of mbl-1"

7. The y-axis scales of figure S3C and S3H should be mended as a full 100% scale.

8. Figure S7E: Statistics should be carefully checked again, as the data of the WT and mec-7 or mec-12 mutants look quite similar at the first glance.

Reviewer #3: Puri et al have submitted a revised version of their manuscript. They have addressed all of my comments well and the manuscript is an improvement overall.

**Have all data underlying the figures and results presented in the manuscript been provided?**

Reviewer #1: Yes

Reviewer #2: Yes

Reviewer #3: Yes

PLOS authors have the option to publish the peer review history of their article (what does this mean?). If published, this will include your full peer review and any attached files.

Reviewer #1: No

Reviewer #2: No

Reviewer #3: No

---

## [Editor Report · Decision Letter 2]

26 Jul 2023

Dear Dr Ghosh-Roy,

Thank you for your revised manuscript that addresses well the reviewers’ criticisms.  We are pleased to inform you that your paper has been accepted.  Attached, please find a marked-up version of the manuscript in which I (MAH) have corrected several grammatical errors/typos.  Please incorporate those corrections as you prepare your final draft for the production team (the editorial team will not need to re-evaluate).  Congratulations on the acceptance of your paper, and thank you for choosing PLOS Genetics as a venue for your work!

Before your submission can be formally accepted and sent to production you will also need to complete our formatting changes, which you will receive in a follow up email. Please be aware that it may take several days for you to receive this email; during this time no action is required by you. Please note: the accept date on your published article will reflect the date of this provisional acceptance, but your manuscript will not be scheduled for publication until the required changes have been made.

Yours sincerely,

Massimo A. Hilliard

Guest Editor

PLOS Genetics

Gregory P. Copenhaver

Editor-in-Chief

PLOS Genetics

Comments from the reviewers (if applicable):

**Data Deposition**

http://datadryad.org/submit?journalID=pgenetics&manu=PGENETICS-D-22-01039R2

**Press Queries**

---

## [Editor Report · Acceptance letter]

15 Aug 2023

PGENETICS-D-22-01039R2 

Muscleblind-1 interacts with tubulin mRNAs to regulate the microtubule cytoskeleton in C. elegans mechanosensory neurons 

Dear Dr Ghosh-Roy, 

We are pleased to inform you that your manuscript entitled "Muscleblind-1 interacts with tubulin mRNAs to regulate the microtubule cytoskeleton in C. elegans mechanosensory neurons" has been formally accepted for publication in PLOS Genetics! Your manuscript is now with our production department and you will be notified of the publication date in due course.

With kind regards,

Zsofi Zombor

PLOS Genetics

On behalf of:
